# Accelerated polymerization of N-carboxyanhydrides catalyzed by crown ether

Yingchun Xia [1,2,8], Ziyuan Song [1,8], Zhengzhong Tan [1], Tianrui Xue[3], Shiqi Wei [4], Lingyang Zhu [5], Yingfeng Yang[1], Hailin Fu[6], Yunjiang Jiang [1], Yao Lin [6], Yanbing Lu [2✉], Andrew L. Ferguson [7✉] & Jianjun Cheng [1,3✉]

The recent advances in accelerated polymerization of N-carboxyanhydrides (NCAs) enriched the toolbox to prepare well-defined polypeptide materials. Herein we report the use of crown ether (CE) to catalyze the polymerization of NCA initiated by conventional primary amine initiators in solvents with low polarity and low hydrogen-bonding ability. The cyclic structure of the CE played a crucial role in the catalysis, with 18-crown-6 enabling the fastest polymerization kinetics. The fast polymerization kinetics outpaced common side reactions, enabling the preparation of well-defined polypeptides using an α-helical macroinitiator. Experimental results as well as the simulation methods suggested that CE changed the binding geometry between NCA and propagating amino chain-end, which promoted the molecular interactions and lowered the activation energy for ring-opening reactions of NCAs. This work not only provides an efficient strategy to prepare well-defined polypeptides with functionalized C-termini, but also guides the design of catalysts for NCA polymerization.

[1] Department of Materials Science and Engineering, University of Illinois at Urbana-Champaign, Urbana, IL 61801, USA. [2] Institute of Polymer Science, College of Chemistry & Chemical Engineering, Hunan University, 410082 Changsha, China. [3] Department of Chemistry, University of Illinois at Urbana-Champaign, Urbana, IL 61801, USA. [4] Department of Bioengineering, University of Illinois at Urbana-Champaign, Urbana, IL 61801, USA. [5] NMR Laboratory, School of Chemical Sciences, University of Illinois at Urbana-Champaign, Urbana, IL 61801, USA. [6] Department of Chemistry and Polymer Program, Institute of Materials Science, University of Connecticut, Storrs, CT 06269, USA. [7] Pritzker School of Molecular Engineering, University of Chicago, Chicago, IL 60637, USA. [8] These authors contributed equally: Yingchun Xia, Ziyuan Song. ✉email: yanbinglu@hnu.edu.cn; andrewferguson@uchicago.edu; jianjunc@illinois.edu

Synthetic polypeptides from ring-opening polymerization (ROP) of N-carboxyanhydrides (NCAs) are a class of promising materials that exhibit interesting self-assembly behaviors and demonstrate unique performance in biomedical applications[1–10]. Despite the great potential of polypeptide materials, conventional polymerization of NCAs initiated with primary amines in N,N-dimethylformamide (DMF) usually takes hours to days to finish, thus suffering from various side reactions during the elongated polymerization time[11], including chain terminations, chain transfers, and water-induced NCA degradations. The last two decades witnessed the advances in controlled ROP of NCAs[12–17], which enabled the preparation of polypeptides with predictable molecular weights (MWs) and narrow dispersity. Nevertheless, the relatively long polymerization time is still a concern, which makes the polymerization susceptible to side reactions and limits the preparation of well-defined and high MW (degree of polymerization, DP > 400) polypeptide materials.

Recently, the developments of accelerated polymerization of NCAs offered great opportunities to not only simplify the preparation process of polypeptides, but also access polymer materials that were difficult to prepare using conventional methods[18–29]. The fast polymerization kinetics outpaces various side reactions, enabling the polymerization in the presence of moisture and even aqueous phase[21,25,27–29]. Well-defined polypeptides with complex structures[19,20,23], high MWs[20,22,26], and unique block sequences[27] were therefore synthesized in an efficient manner, which is difficult, if not impossible, with conventional NCA ROP methods.

We recently reported the covalent, cooperative polymerization (CCP) of NCAs in solvents with low dielectric constants[20,24,25,27,28]. The formation of α-helical structures and the close proximity between initiating sites catalyzed the growth of polypeptides, resulting in the fast consumption of NCA monomers. Nevertheless, the linear polymerization from primary amines, the conventional initiators for NCA polymerization, is still relatively slow, especially at low monomer concentrations[25,28]. Considering that the binding between propagating polypeptide chains and NCA monomers is crucial for the self-catalysis behaviors[28], we reasoned that a catalyst promoting amine/NCA interactions would further accelerate the polymerization, enabling the rapid and efficient synthesis of polypeptide materials.

Herein we report the accelerated polymerization of NCAs from various primary amine initiators, which is catalyzed by the addition of crown ether (CE) in solvents with low polarity and low hydrogen-bonding ability. The ring structure and the size of CE play an important role in the rate acceleration, with the polymerization in the presence of 18-crown-6 (18-C-6) exhibiting the fastest rate. Polypeptides with tunable MWs, narrow dispersity, and functionalized C-terminus were prepared in a rapid manner, which outpaces various side reactions including chain transfers and water-induced NCA degradations. We believe this work will enrich the toolbox of accelerated polymerization of NCAs, paving the way for the rapid synthesis of well-defined polypeptides.

## Results and discussion

**CE-catalyzed, fast polymerization of NCAs.** It has been reported that CE facilitated the reactions between primary amines and electrophiles, depending on the solvent selection, as well as the size of the ring molecule[30]. Inspired by this result, we hypothesized that CE could also catalyze the NCA polymerization in a proper solvent, which involves the ring-opening reaction of NCA by reacting with amines. To demonstrate the catalytic activity of CE, 18-C-6 was first mixed with γ-benzyl-L-glutamate NCA (BLG-NCA) in dichloromethane (DCM) in 1:100 molar ratio,

into which n-hexylamine (Hex-NH$_2$) was added to initiate the polymerization ([M]$_0$ = 50 mM, [M]$_0$/[I]$_0$ = 100) (Fig. 1a, b). Fourier transform infrared spectroscopy (FTIR) analysis revealed a fast decrease in anhydride peaks (1865 and 1793 cm$^{-1}$) from BLG-NCA (Supplementary Fig. 1a), reaching 95% conversion within 18 min (Fig. 1c). The increase in amide peaks at 1652 and 1550 cm$^{-1}$ suggested the formation of α-helical poly(γ-benzyl-L-glutamate) (PBLG), which is consistent with the previously reported CCP of NCAs in DCM[20,24,28]. The polymerization kinetics was also monitored by nuclear magnetic resonance (NMR), which revealed the rapid disappearance of proton signals from BLG-NCA and the appearance of proton signals from PBLG (Supplementary Fig. 1b). In sharp contrast, the polymerization in the absence of CE under identical conditions was much slower, with 75% conversion observed after 12 h (Fig. 1c), demonstrating the remarkable catalytic activity of CE. Additionally, with the same batch of NCA, CE-catalyzed polymerization in DCM was much faster than conventional polymerization systems, including bipyNi(COD) initiating system in tetrahydrofuran (THF) and hexamethyldisilazane (HMDS) initiating system in DMF (Supplementary Fig. 1c). The increase in [M]$_0$ further shortened the polymerization time, where >95% conversion of NCA was observed within 2 min at [M]$_0$ = 400 mM (Supplementary Fig. 1d). Vigorous evolution of bubbles was observed at high [M]$_0$ (Fig. 1d and Supplementary Movie 1), suggesting the fast release of CO$_2$. In order to rule out the possibility that CE served as an initiator for the fast polymerization process, NCA was mixed with 18-C-6 in CD$_2$Cl$_2$ under anhydrous conditions, which was stable for at least 40 min as evidenced by $^1$H NMR (Supplementary Fig. 2a). Additionally, CE was introduced at different time points during the Hex-NH$_2$-initiated polymerization of BLG-NCA. The monomer was rapidly consumed upon the addition of CE in all cases (Supplementary Fig. 2b), excluding the possibility that CE served as a co-initiator.

We have reported previously that the polymerization of NCA in DCM followed a two-stage, CCP behavior, where the rate constant for the second stage ($k_2$) was much larger than that of the first stage ($k_1$) ($k_2/k_1$ = 10–10,000)[20,24,28]. The rate difference of the two stages resulted in the formation of both long polymers and short oligomers, with the MWs of polymers larger than the expected values from [M]$_0$/[I]$_0$. The obtained PBLG from CE-catalyzed polymerization was characterized by gel permeation chromatography (GPC), with the obtained MW (136 kDa) much larger than the expected MW (22 kDa) (Supplementary Fig. 3a, b), suggesting the cooperative polymerization behavior[31]. The existence of a slow, first stage during CE-catalyzed polymerization was verified by circular dichroism (CD), where a slow increase in CD signal at 222 nm was observed, corresponding to the random-coiled conformation of propagating polypeptides (Supplementary Fig. 3c)[20]. While the increase in [CE]$_0$/[I]$_0$ ratio did not significantly change the rate acceleration behavior (Supplementary Fig. 4a), the decrease in [CE]$_0$/[I]$_0$ to 1:20 slowed down the polymerization kinetics, with 74% conversion after 6 h that is still much faster than that in the absence of CE (Supplementary Fig. 4b). In addition, while CE catalyzed the polymerization of racemic γ-benzyl-DL-glutamate NCA (BDLG-NCA) (Supplementary Fig. 4c), the catalyzed polymerization was much slower than that of BLG-NCA, with 50% monomer conversion after 3 h (Fig. 1e), suggesting the importance of ordered α-helical structure for the accelerated polymerization[20,24]. As a comparison, the polymerization of γ-benzyl-D-glutamate NCA (BDG-NCA) showed a comparable rate with that of BLG-NCA (Fig. 1e), indicating a negligible effect of helix sense on the rate enhancement.

Motivated by the fast polymerization catalyzed by 18-C-6, we went on to study the CE-catalyzed polymerization with different primary amine initiators, aiming to synthesize versatile polypeptide

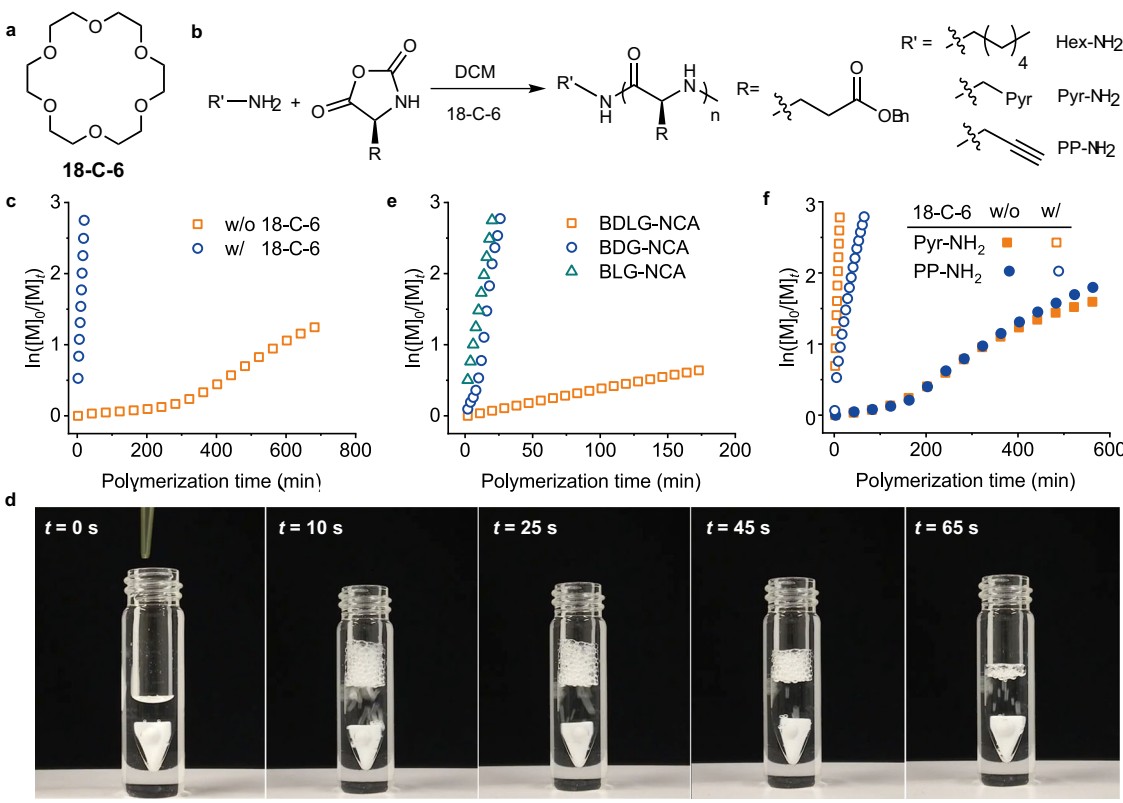

**Fig. 1 CE-induced accelerated polymerization of NCA. a** Chemical structure of 18-C-6. **b** Scheme illustrating CE-catalyzed polymerization of NCA from small-molecular amine initiators. **c** Semilogarithmic kinetic plot of polymerization of BLG-NCA in DCM initiated by Hex-NH$_2$ in the presence and absence of 18-C-6. $[M]_0/[I]_0 = 100$, $[I]_0 = [CE]_0 = 0.5$ mM. **d** Snapshots showing the fast consumption of BLG-NCA accompanied with the rapid evolution of carbon dioxide at $[M]_0/[I]_0 = 100$, $[I]_0 = [CE]_0 = 4$ mM. **e** Semilogarithmic kinetic plot of polymerization of BLG-NCA, BDG-NCA, and BDLG-NCA in DCM initiated by Hex-NH$_2$ in the presence of 18-C-6. $[M]_0/[I]_0 = 100$, $[I]_0 = [CE]_0 = 0.5$ mM. **f** Semilogarithmic kinetic plot of polymerization of BLG-NCA in DCM initiated by various small-molecular initiators in the presence and absence of 18-C-6. $[M]_0/[I]_0 = 100$, $[I]_0 = [CE]_0 = 0.5$ mM.

materials with functionalized C-terminus. The addition of 18-C-6 boosted the polymerization of BLG-NCA initiated from various primary amines in DCM, shortening the polymerization time from >10 h to <60 min (Fig. 1f and Supplementary Fig. 5a, b). Various functionalities, including a pyrenyl group (Pyr) as a fluorescence probe and a propargyl group (PP) for post-polymerization modification, can be easily incorporated at the C-terminus of polypeptides by selecting the corresponding primary amines as initiators (Supplementary Fig. 5c). By stopping the polymerization at the early stage, we successfully confirmed the incorporation of these functional groups through NMR and matrix-assisted laser desorption/ionization-time of flight mass spectrometry (MALDI-TOF MS) (Supplementary Fig. 6).

**Impact of catalyst size and structure**. In an attempt to further elucidate the acceleration factors, we tested the catalytic activity of several small-molecular or polymeric analogs of 18-C-6 bearing ethyleneoxy (–CH$_2$CH$_2$O–) moieties (Fig. 2a, b). All polymerizations in the presence of cyclic CEs showed accelerated, two-stage, cooperative behavior in DCM, with 18-C-6 exhibiting the highest catalytic activity (Fig. 2c). The polymerizations in the presence of 24-crown-8 (24-C-8) with a larger ring and 15-crown-5 (15-C-5) with a smaller ring were slower compared with 18-C-6, requiring 110 min and 180 min to reach >90% NCA consumption, respectively. This result is consistent with the previously reported CE-size-dependent rotaxane formation[30], presumably because 18-C-6 provided the best conformation to catalyze the ring-opening reaction. The polymerization with the

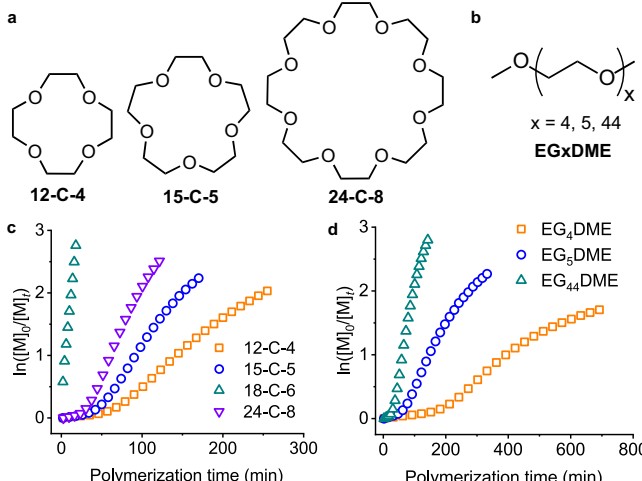

**Fig. 2 Structural impact of CE and its analogs on the catalysis. a** Chemical structures of CEs. **b** Chemical structures of LE analogs. **c** Semilogarithmic kinetic plot of polymerization of BLG-NCA in DCM initiated by Hex-NH$_2$ in the presence of various CEs. **d** Semilogarithmic kinetic plot of polymerization of BLG-NCA in DCM initiated by Hex-NH$_2$ in the presence of various LE analogs. $[M]_0/[I]_0 = 100$, $[I]_0 = [CE/LE]_0 = 0.5$ mM.

least efficient catalyst, 12-crown-4 (12-C-4), on the other hand, took 5 h to achieve the same monomer conversion.

The ring structure of CEs is crucial for fast polymerization kinetics, as the linear ether (LE) analogs showed much lower

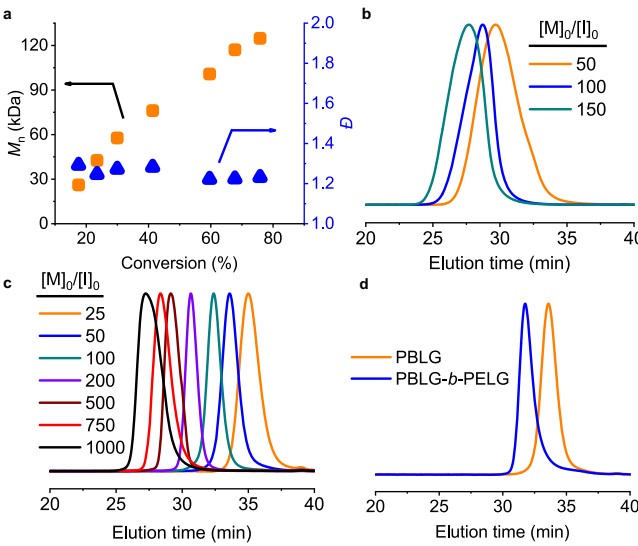

**Fig. 3 MW control for CE-catalyzed polymerization of NCA. a** The obtained MW and dispersity at various conversions during CE-catalyzed polymerization of BLG-NCA initiated by Hex-NH$_2$ in DCM. $[M]_0/[I]_0 = 100$, $[I]_0 = [CE] = 0.5$ mM. **b** GPC-LS traces of the obtained PBLG initiated by Hex-NH$_2$ in the presence of 18-C-6. $[I]_0 = [CE]_0 = 0.5$ mM. **c** Normalized GPC-LS traces of the obtained PBLG initiated by α-helical PBLG macroinitiator in the presence of 18-8-6. $[M]_0 = 100$ mM, $[I]_0 = [CE]_0$. **d** Normalized GPC-LS traces showing the synthesis of block copolypeptides initiated by α-helical PBLG macroinitiator in the presence of 18-C-6. $[M]_0/[I]_0 = 50$, $[I]_0 = [CE]_0 = 1$ mM and 0.5 mM for the first and second block, respectively.

activity. For instance, only 14% of BLG-NCA was converted after 3-h polymerization in the presence of tetraethylene glycol dimethyl ether (EG$_4$DME), the linear analog of 15-C-5 (Fig. 2d). Extending the ethyleneoxy units resulted in faster polymerization; the polymerization catalyzed by pentaethylene glycol dimethyl ether (EG$_5$DME) exhibited a comparable rate with that by 15-C-5 (Fig. 2d). Further elongating the –CH$_2$CH$_2$O– units into a polymeric catalyst, poly(ethylene glycol) dimethyl ether (2 kDa, EG$_{44}$DME), led to even faster polymerization that reached 90% conversion after 110 min (Fig. 2d). Nevertheless, the polymerization catalyzed by linear, polymeric EG$_{44}$DME was still slower than the top-performing cyclic CE, 18-C-6, likely due to the less well-suited conformation with the flexible chains.

**Control over MWs**. In order to check the livingness of CE-catalyzed polymerization of NCA initiated by Hex-NH$_2$, we stopped the polymerization at different conversions, and characterized the obtained polypeptides by GPC. The obtained MW increased linearly with the higher conversion of NCA (Fig. 3a), suggesting the maintained end-group fidelity (i.e., propagating primary amines) throughout the polymerization process. While the CE-catalyzed polymerization produced polypeptides with much larger MW than expected, we were still able to tune the MWs by changing the feeding $[M]_0/[I]_0$ (Fig. 3b and Table 1). The apparent initiation efficiency (IE, defined as the ratio of expected MW to obtained MW) was calculated to be 15–25%, with higher IE observed at higher $[I]_0$ or higher $[M]_0/[I]_0$.

The low IE value of CE-catalyzed polymerization originated from the rate differences between the two stages of the cooperative polymerizations[31]. In order to solve this issue and provide better control over MWs and dispersity, we prepared α-helical PBLG macroinitiators to skip the slow, first stage kinetics (Supplementary Fig. 7)[25]. As expected, the use of helical PBLG

macroinitiators resulted in a fast, one-stage polymerization kinetics, generating well-defined polypeptides with predictable MWs and narrow dispersity ($Đ = M_w/M_n$, <1.1) (Fig. 3c and Table 2). The CE-catalyzed polymerization was also applied to other NCA monomers, including γ-ethyl-ʟ-glutamate NCA (ELG-NCA), γ-(4-propargyloxy)benzyl-ʟ-glutamate NCA (POB-NCA), and $N^{\varepsilon}$-carboxybenzyl-ʟ-lysine NCA (ZLL-NCA), all leading to well-defined polypeptides (Supplementary Fig. 8), demonstrating the universal catalytic activity of 18-C-6 for the preparation of versatile polypeptide materials. The living nature of CE-catalyzed polymerization enabled us to prepare well-defined block copolypeptides in a short time through the sequential addition of NCA monomers (Fig. 3d and Table 2). In addition, Pyr-NH$_2$ and PP-NH$_2$ were used to prepare C-terminus functionalized PBLG macroinitiators, which enabled the preparation of functionalized polypeptides with predictable MWs in the presence of CE (Supplementary Fig. 9).

**Minimal side reactions by CE-catalyzed, accelerated polymerization**. Various side reactions occur during NCA polymerization, which resulted in poorly controlled polypeptide structures. The accelerated polymerization kinetics outpaces these side reactions,

## Table 1 Accelerated polymerization of NCA initiated from Hex-NH$_2$ catalyzed by CE.

| Entry[a] | $[I]_0$/mM | $[M]_0/[I]_0$ | $t$/min[b] | $M_n/M_n{}^{\star}$(kDa)[c,d] | $Đ$[d] |
|---|---|---|---|---|---|
| 1 | 0.5 | 50 | 12 | 76.4/11.1 | 1.24 |
| 2 | 0.5 | 100 | 18 | 136/22.0 | 1.21 |
| 3 | 0.5 | 150 | 25 | 206/33.0 | 1.28 |
| 4 | 2 | 100 | 5 | 107/22.0 | 1.28 |
| 5 | 3 | 100 | 3 | 100/22.0 | 1.34 |
| 6 | 4 | 100 | 2 | 88.7/22.0 | 1.33 |

[a]All polymerizations were conducted at room temperature in DCM using Hex-NH$_2$ as the initiator, BLG-NCA as the monomer, and 18-C-6 as catalyst. $[I]_0 = [18\text{-}C\text{-}6]_0$.
[b]Polymerization time reaching 95% monomer conversion.
[c]Obtained MWs/Designed MWs*.
[d]Determined by GPC; $dn/dc = 0.100$–0.105.

## Table 2 Accelerated polymerization of NCA initiated by helical PBLG macroinitiator catalyzed by CE.

| Entry[a] | Monomer[b] | $[M]_0/[I]_0$ | $t$/min[c] | $M_n/M_n{}^{\star}$(kDa)[d,e] | $Đ$[e] |
|---|---|---|---|---|---|
| 1 | BLG | 25 | 5 | 11.2/12.7 | 1.05 |
| 2 | BLG | 50 | 14 | 19.7/18.3 | 1.06 |
| 3 | BLG | 100 | 19 | 32.6/29.2 | 1.05 |
| 4 | BLG | 200 | 21 | 56.1/51.2 | 1.05 |
| 5 | BLG | 500 | 45 | 119/117 | 1.05 |
| 6 | BLG | 750 | 150 | 166/172 | 1.06 |
| 7 | BLG | 1000 | 270 | 230/226 | 1.09 |
| 8 | ELG | 100 | 110 | 24.5/22.9 | 1.06 |
| 9 | POB | 100 | 41 | 41.0/34.5 | 1.09 |
| 10 | ZLL | 100 | 25 | 35.7/33.4 | 1.05 |
| 11[f] | BLG/ELG | 50 + 50 | 14 + 100 | 31.1/26.0 | 1.06 |

[a]All polymerizations were conducted at room temperature in DCM using α-helical PBLG as the macroinitiator and 18-C-6 as the catalyst. $[M]_0 = 100$ mM, $[I]_0 = [CE]_0$.
[b]BLG γ-benzyl-ʟ-glutamate NCA, ELG γ-ethyl-ʟ-glutamate NCA, POB γ-(4-propargyloxy)benzyl-ʟ-glutamate NCA, ZLL N$\varepsilon$-carboxybenzyl-ʟ-lysine NCA.
[c]Polymerization time reaching 95% monomer conversion.
[d]Obtained MWs/Designed MWs*.
[e]Determined by GPC; $dn/dc = 0.093$–0.119 except for Entry 8 and Entry 11 (i.e., polypeptides containing ELG residues), which has a $dn/dc$ value of 0.072 and 0.088, respectively.
[f]Block copolymerization by the sequential addition of monomers.

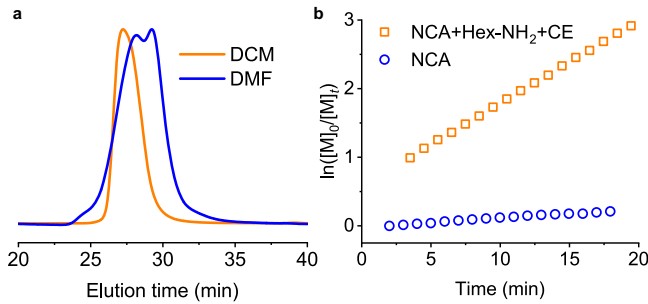

**Fig. 4 Minimal side reactions with CE-catalyzed polymerization of NCA.**
**a** Normalized GPC-LS traces of the obtained PBLG in DCM and DMF initiated by α-helical PBLG macroinitiator in the presence of 18-C-6. $[M]_0/[I]_0 = 1000$, $[M]_0 = 100$ and 400 mM for the polymerization in DCM and DMF, respectively. **b** Semilogarithmic kinetic plot of polymerization of BLG-NCA in a biphasic system in the presence and absence of Hex-NH$_2$ and CE. $[M]_0/[I]_0 = 100$, $[I]_0 = [CE]_0 = 0.5$ mM, DCM:water = 100:1 (w/w).

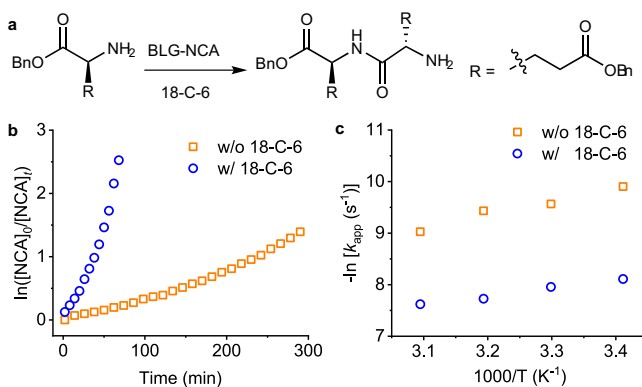

**Fig. 5 Ring-opening reaction of NCA catalyzed by CE. a** Scheme illustrating CE-catalyzed ring-opening reaction of BLG-NCA by DBLG.
**b** Semilogarithmic kinetic plot of ring-opening reaction of BLG-NCA in DCM by DBLG in the presence and absence of 18-C-6 at room temperature by FTIR. **c** Arrhenius plot of ring-opening reaction of BLG-NCA by DBLG in CDCl$_3$ in the absence and presence of 18-C-6. $[DBLG]_0 = 30$ mM, $[NCA]_0 = [CE]_0 = 3$ mM.

allowing us to prepare well-defined polypeptides[8]. For instance, conventional polymerization of BLG-NCA initiated by PBLG macroinitiators in DMF at $[M]_0/[I]_0 = 1000$ resulted in a polypeptide with a bimodal molecular-weight distribution (MWD) ($Đ = 1.35$) after 72 h (Fig. 4a and Supplementary Table 1), which is attributed to significant chain transfers[11]. As a comparison, the CE-catalyzed preparation of PBLG at $[M]_0/[I]_0 = 1,000$ was successfully prepared within 4.5 h. The obtained MW (230 kDa) agreed well with the designed MW (226 kDa), with 95% end-group retained after polymerization for further chain extensions or post-polymerization modifications. In addition, CE-catalyzed polymerization exhibited good resistance to water-induced degradation of NCAs due to the fast consumption of NCAs. Under the catalysis of 18-C-6, the polypeptides obtained in normal, non-dried DCM showed similar MW and dispersity compared with that obtained from anhydrous DCM (Supplementary Fig. 10a and Supplementary Table 2).

The fast polymerization kinetics and the moisture-tolerance of CE-catalyzed polymerization encouraged us to prepare polypeptide materials directly from non-purified NCAs[25]. To validate that CE-catalyzed polymerization in the presence of aqueous phase, the polymerization of purified BLG-NCA was first conducted in a DCM/water biphasic system in the presence of 18-C-6. The NCA monomer reached >95% conversion after

20 min, which is much faster than the water-induced degradation under similar conditions (Fig. 4b), suggesting minimal impact of the aqueous phase on the CE-catalyzed polymerization. Since the accelerated polymerization kinetics outpaced water-induced degradation of NCAs, we were able to introduce the aqueous phase for the removal of acidic and electrophilic impurities in non-purified NCAs in situ, which would otherwise quench the amino groups and inhibit the polymerization. Moreover, different from other initiators/catalysts that enable accelerated polymerization of NCAs, 18-C-6 exhibited excellent water-stability. Therefore, we prepared the non-purified BLG-NCA by skipping the purification steps, which was then washed with aqueous buffer and polymerized by the addition of a DCM solution of CE and Hex-NH$_2$. Well-defined homopolypeptides were prepared within 30 min (Supplementary Fig. 10b), highlighting the usefulness of CE-catalyzed polymerizations.

**Ring-opening reaction of NCA catalyzed by CE.** The fast polymerization kinetics and the associated benefits provide an impetus for us to elucidate the mechanism of the CE-catalyzed rate acceleration. In order to simplify the study, α,γ-dibenzyl-L-glutamate (DBLG) was selected to mimic the N-terminus of polypeptides during the chain propagation step (Fig. 5a), which was used to open the BLG-NCA ring in the presence and absence of CE. DBLG has the same α-substitutions of amino group with the propagating PBLG chain, which serves as an ideal small-molecular compound to mimic the local bulkiness of the propagating polypeptide terminus. The $[DBLG]_0/[NCA]_0$ ratio was set to 5–15 to minimize the polymerization of NCAs, reducing unnecessary complexity for the analysis. As shown in Fig. 5b, the addition of 18-C-6 catalyzed the ring-opening reaction of NCA, reaching full conversion after 80 min. As a comparison, only ~22% was consumed under identical conditions in the absence of 18-C-6. By testing the reaction rate at various concentrations of 18-C-6 in CDCl$_3$ with NMR, we determined that the reaction order is 0.26 with respect to CE (see Supplementary Fig. 11 for details). This result agrees well with the kinetics studies (Supplementary Fig. 4a, b), where reducing the amount of 18-C-6 slowed down the polymerization rate.

The ring-opening reaction of BLG-NCA by DBLG was also studied using NMR in CDCl$_3$ at various temperatures. The apparent rate constant, $k_{app}$, was quantified by monitoring the decrease in the NMR signals at 4.53-4.15 ppm (i.e., the α-H of BLG-NCA, chemical shift changes depending on the concentration and temperature), which became larger at a higher temperature. The Arrhenius plot showed a linear relationship between $-\ln(k_{app})$ and $1/T$ (Fig. 5c), where the apparent activation energy can be obtained from the slope. The activation energy was calculated to be $(13.4 \pm 1.1)$ and $(21.7 \pm 2.9)$ kJ/mol in the presence and absence of 18-C-6, respectively.

**Enhanced polypeptide/NCA interactions by CE.** We have reported recently that the binding interaction between NCA and propagating polypeptide played a crucial role in the rate acceleration in CCP of NCA in DCM[28]. Therefore, we hypothesized that CE participated in and promoted the polypeptide/NCA binding, facilitating the ring-opening reaction of NCAs. To test our hypothesis, we first studied the CE-catalyzed polymerization kinetics in various solvents. The selection of solvent plays an important role in the cooperative polymerization of NCAs, as the solvent molecule may disrupt the binding interactions and slow down the kinetics[20,24,28]. While similar CE-catalyzed acceleration was observed in chloroform, the polymerization takes much longer time to finish in polar or hydrogen-bonding solvents like DMF and THF, even in the presence of 18-C-6 (Supplementary

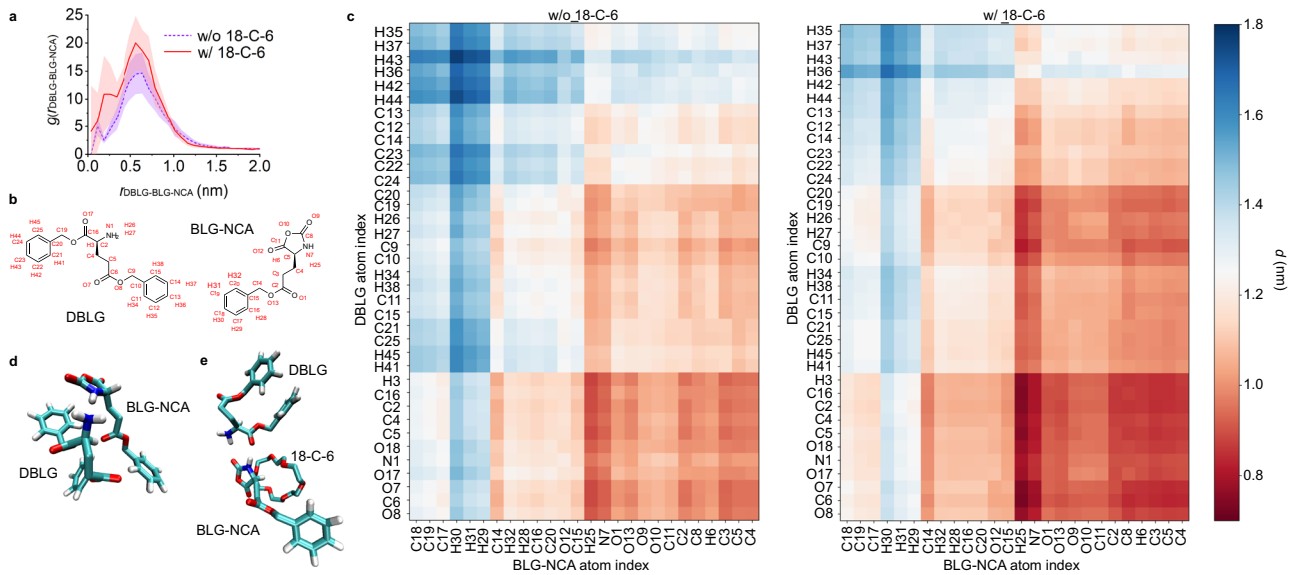

**Fig. 6 Structural characterization of BLG-NCA (3 mM) and DBLG (30 mM) approach in the presence and absence of 18-C-6 (3 mM). a** Radial distribution functions g($r_{DBLG-BLG-NCA}$) in the center-of-geometry distance between BLG-NCA and DBLG molecules in the presence (red curve) and absence (blue curve) of 18-C-6. The shaded areas correspond to the estimated uncertainties in the curves computed by partitioning the production run into five equal sized blocks and computing standard deviations over the blocks. The presence of CE increases the number of closely associated DBLG-NCA pairs, producing a 27% enhancement in number density of BLG-NCA/DBLG pairs within a spherical volume of radius $r = 1$ nm. **b** Scheme labeling the atom indices used for the contact matrices. **c** Contact matrices for the interatomic distances between the atoms comprising BLG-NCA and DBLG in the absence and presence of 18-C-6. Interatomic pairwise distances are averaged over all BLG-NCA/DBLG contact pairs with a center-of-geometry distance $r_{DBLG-BLG-NCA} < 1$ nm, with 5774 such configurations observed in the absence of 18-C-6 and 7129 configurations in its presence. Standard errors in the computed distances are estimated by block averaging of the trajectory into five equally sized contiguous blocks and in all cases are smaller than ±0.15 nm. The presence of CE changes the geometry of approach within the BLG-NCA/DBLG contact pairs. The most prominent effect is to induce the red sector in the bottom right of the contact matrix to become redder, indicating that the atoms participating in these contacts get closer. **d**, **e** Snapshots of a BLG-NCA and DBLG contact pair in a bimolecular complex and a termolecular complex comprising a mediating 18-C-6 molecule, BLG-NCA and DBLG. C = teal, H = white, O = red, N = blue.

Fig. 12a). DMF and THF are solvents with strong hydrogen bonding (H-bonding) accepting ability[32], which likely disrupt the H-bonding interactions between CE and propagating amine/NCA. In addition, the addition of 18-C-6 exhibited insignificant acceleration of HMDS-initiated polymerization of BLG-NCA (Supplementary Fig. 12b), presumably due to the existence of bulky N-trimethylsilyl carbamate at the chain terminus during chain propagation[16], which blocked the interactions with CE.

In order to further elucidate the catalysis mechanism, we first studied the bimolecular binding interactions of CE/amine-terminated PBLG (PBLG-NH$_2$) and CE/NCA through diffusion-ordered spectroscopy (DOSY) (Supplementary Fig. 13a). The addition of PBLG-NH$_2$ into the CD$_2$Cl$_2$ solution of 18-C-6 resulted in an obvious decrease in its diffusion coefficient by 20% (Supplementary Fig. 13b), indicating molecular interactions between PBLG-NH$_2$ and 18-C-6 in DCM. In contrast, the addition of PBLG with acetyl-capped N-terminus, PBLG-NHAc, led to negligible changes in the diffusion coefficient of 18-C-6 (Supplementary Fig. 13b), suggesting that the binding interactions occur at the N-terminus of PBLG-NH$_2$[28]. In addition, the mixing of 18-C-6 and BLG-NCA led to a decrease in diffusion coefficients of both molecules (Supplementary Fig. 13c, d), confirming the binding interactions between CE and NCA. In an attempt to directly probe the termolecular interactions, we collected the DOSY spectra of CE/PBLG-NH$_2$/NCA at low temperature to slow down the reaction. Upon the mixing of BLG-NCA with CE/PBLG-NH$_2$ mixture, a decrease in the diffusion coefficient of BLG-NCA was observed (Supplementary Fig. 13e), validating the interaction of NCA with the PBLG-NH$_2$/CE complexes. These DOSY experiments collectively suggested that CE was able to bind with both propagating polypeptides and NCA,

resulting in enhanced polypeptide/NCA interactions. Since the addition of C$^5$=O of NCA ring by NH$_2$ group of propagating amine is the rate-limiting step during NCA polymerization[33], CE-promoted interactions between NCA and propagating polypeptide likely lower the activation energy and stabilize the transition state, leading to the rate acceleration of ROP.

Besides DOSY experiments, the binding between CE and NCA was further demonstrated by NMR titration studies. Addition of 18-C-6 resulted in significant downfield shift of ring N–H signal from BLG-NCA in CD$_2$Cl$_2$ (Supplementary Fig. 14a), indicating H-bonding interactions between BLG-NCA and 18-C-6 that agree well with the DOSY results. By plotting the changes in [NCA]*Δδ against the molar fraction of NCA with the continuous variation method (i.e., the Job plot)[34], we were able to calculate the binding stoichiometry between NCA and CE to be 2:1 (Supplementary Fig. 14b). A proposed scheme illustrating the catalytic role of CE during NCA polymerization was included in Supplementary Fig. 15.

**Molecular dynamics simulation.** To further probe the acceleration mechanism, we conducted molecular dynamics simulations of DBLG at 30 mM and BLG-NCA at 3 mM in DCM in the presence and absence of 18-C-6 at 3 mM (Supplementary Movies 2 and 3). The presence of CE in a 1:1 ratio with BLG-NCA was observed to induce more frequent and closer associations between DBLG and BLG-NCA. We quantified this in the radial distribution function in the DBLG to BLG-NCA center-of-geometry, which exhibits a 27% increase in number density of BLG-NCA/DBLG pairs within a spherical volume of radius $r = 1$ nm in the presence of CE (Fig. 6a). This enhancement in

interactions is consistent with experimental results showing an elevated rate of NCA ring-opening reaction in the presence of CE (Fig. 5b).

The presence of 18-C-6 also modified the interaction geometry within BLG-NCA/DBLG contact pairs as revealed by changes in the interatomic contact matrices (Fig. 6b, c). The most prominent difference in the contact matrices is a ~25% reduction in the mean distance of approach between the ring N–H in the BLG-NCA ring and the α-carbonyl group in DBLG (i.e., the N7-H25 and O17 pair in Fig. 6c) from an average value of ~1 nm to ~0.75 nm in the presence of CE. Representative snapshots of BLG-NCA/DBLG contact pairs show that this change can be explained in part by the participation of CE within BLG-NCA/DBLG contact pairs through the formation of tertiary molecular complexes, in which the CE mediates the BLG-NCA/DBLG interaction (Fig. 6d, e and Supplementary Fig. 16). We propose that the enhanced proximity of these groups promotes a favorable geometry for NCA ring-opening reaction and suggests a molecular mechanism for the catalytic performance of CE.

In summary, we reported the CE-catalyzed, accelerated polymerization of NCAs in solvents with low polarity and low hydrogen-bonding ability. CE promoted the interactions between propagating amino groups and NCA monomers, resulting in fast polymerization kinetics. The rapid polymerization catalyzed by CE enabled us to prepare well-defined polypeptide materials in an efficient manner with minimal side reactions, including chain transfers and water-induced NCA degradations. This work not only sheds lights on the mechanism of cooperative polymerization of NCA, but also facilitates the preparation of polypeptide materials from primary amine initiators.

## Methods

**Polymerization kinetics.** The DCM or $CD_2Cl_2$ solution of CE and NCA were mixed, followed by the addition of small molecular or macromolecular amine initiators. The resulting solution was transferred into a FTIR cell or an NMR tube to monitor the kinetics. The resulting solution was dried under vacuum and injected into GPC without further purification (DMF containing 0.1 M LiBr as the mobile phase).

**Polymerization of NCAs in a biphasic system.** The DCM solution of purified or non-purified BLG-NCA was mixed with aqueous buffer (pH = 9.0) and vigorously vortexed for 10 s, into which the DCM solution of initiator and 18-C-6 was added.

**Determination of activation energy.** The $CDCl_3$ solution of DBLG and BLG-NCA were mixed in the presence or absence of 18-C-6, and the NMR spectra of the resulting solution were collected at various temperatures. The apparent rate constant ($k_{app}$) was determined from the decrease in signal of the α-H of NCA, whose natural logarithm value was plotted against $1/T$ to calculate the activation energy.

**Diffusion-ordered spectroscopy (DOSY) studies.** The $^1H$ DOSY experiments were performed for the $CD_2Cl_2$ or $CDCl_3$ solution of BLG-NCA, PBLG-NH$_2$, PBLG-NHAc, 18-C-6, or their mixtures by means of a convection compensated gradient stimulated echo pulse sequence (DgsteSL_cc). The spectra were processed with the Agilent VrnmJ4.2A software, and the diffusion coefficients were extracted therein.

**Determination of binding stoichiometry between CE/NCA.** The $CD_2Cl_2$ solution of BLG-NCA and 18-C-6 were mixed at different ratios and the NMR spectra were collected. The change in $[NCA]*\Delta\delta$ was plotted against the molar fraction of NCA to determine the binding stoichiometry.

**Molecular dynamics simulation.** Molecular dynamics simulations were conducted using the Gromacs 2019 simulation suite[35]. Molecular topologies for 18-C-6, DBLG, BLG-NCA, and DCM solvent were constructed using the Automated Topology Builder server (http://atb.uq.edu.au)[36]. Molecular interactions were treated using the Groningen Molecular Simulation (GROMOS) 54A7 force field[37]. Simulation trajectories were analyzed using the MDAnalysis libraries[38,39] and visualized using Visual Molecular Dynamics[40].

## Data availability

The data that support the findings of this study are available within the paper and its Supplementary Information files. Any other data are available from the corresponding authors upon reasonable request. Source data are provided with this paper.

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

## Acknowledgements
This material is based upon work supported by National Science Foundation under Grant Nos. CHE-1709820, CHE-1905097, and DMS-1841810. We thank R. Wang and K. Cai for helpful discussions. Y. X., a visiting student from Hunan University, China, acknowledges the support from China Scholarship Council for her studies in Professor Jianjun Cheng's laboratory at UIUC.

## Author contributions
Y.X., Z.S., T.X., Y.L., A.L.F., and J.C. conceived the idea and designed the experiments. Y.X., Z.S., Z.T., T.X., S.W., L.Z., Y.Y., H.F., Y.J., Y.L., and A.L.F. performed the experiments. Y.X., Z.S., Y.L., A.L.F., and J.C. analyzed data and prepared the manuscript with contributions from all authors.

## Competing interests
The authors declare no competing interests.
