## [Peer Review File · Nature Communications]

Reviewers' comments:

Reviewer #1 (Remarks to the Author):

Review of NCOMMS-20-07513

The authors report a rapid crown ether (CE)-catalyzed ROP of α -amino acid-NCAs in solvents of low dielectric constants, such as DCM, initiated by primary amines. Based on experimental results (kinetics, activation energy, DOSY, binding stoichiometry) and molecular dynamic simulation, they suggest that the presence of CE changes the binding geometry between NCA and propagating amino chain-ends and thus lowers the activation energy, resulting in rapid cooperative ROP.

This MS has a similar philosophy as the two previous papers published by the same groups.

In their paper (Nat. Chem. 2017, 9, 614-622, see Figure below), the authors used a TMS-protected amine brush macroinitiator for the ROP of NCAs. They found an auto-accelerated polymerization in the brush-like macromolecular architecture, where the cooperative interactions between the macrodipoles of neighboring helical polypeptides dramatically accelerate the ROP of NCAs.

Reprinted by permission from Baumgartner, R., Fu, H., Song, Z. et al. Cooperative polymerization of α -helices induced by macromolecular architecture. Nature Chem 9, 614–622 Copyright 2017.

Subsequently, in 2019 (J. Am. Chem. Soc. 2019, 141, 8680-8683, see Figure below), the same groups used linear aliphatic diamines, instead of brush-like macroinitiators, for the ROP of NCAs. They found, similarly, that the “hinged” structure accelerates the ROP dramatically as compared to that initiated by monoamines, such as hexylamine.

Reprinted (adapted) with permission from J. Am. Chem. Soc. 2019, 141, 22, 8680–8683. Copyright 2019 American Chemical Society.

In this MS, the author first used small primary amines to initiate the ROP of NCAs in the presence of crown ethers. The polymerization is fast but uncontrolled, resulting in polypeptides with very high molecular weight than expected and high Đ (1.21-1.33). In order to achieve control over the polymerization, the author used a pre-synthesized helical PBLG (DP=30) as macroinitiator.

The authors mention in the introduction, *“Nevertheless, the linear polymerization from small molecular primary amines, the conventional initiators for NCA polymerization, is still relatively slow, especially at low monomer concentrations. Considering that the binding between propagating polypeptide chains and NCA monomers is crucial for the self-catalysis behaviors, we reasoned that a catalyst promoting amine/NCA interactions would further accelerate the polymerization, enabling the rapid and efficient synthesis of polypeptide materials.”* However, in the presence of crown ether, the polymerization initiated by commercial available small amines is fast but uncontrolled, resulting in polypeptides with higher molecular weight than the expected ones and high Đ (1.21-1.33)."

Consequently, the fast and controlled polymerization can only be achieved by using a pre-synthesized helical PBLG macroinitiator.

A few other comments

-I suggest the author to use $\ln\{[M]_0/[M]_t\}$ vs time instead of conversion vs time plots to better analyze the ROP results.

-Page 14, *“By testing the reaction rate at various concentrations of 18-C-6, we determined that the reaction order is 0.61 in CE (see Supplementary Fig.7 for details)...”* The kinetic study in Figure 7 (SI) is based only on three points. To have more reliable data, more points are needed.

-Figure 3C, the GPC plot, corresponding to the polymerization with $[M]/[I]=1000$ shows a visible shoulder (usually indicating an overlap of two molecular weight distributions). I suggest the author to carry out a polymerization with monomer/initiator ratio between 500 and 1000 to see the gradual changes of molecular weight distribution.

-The authors should include the DOSY spectra in Supplementary Information.

-The authors should include the dn/dc values, used for the molecular weights calculations of the resultant polypeptides, in the note of the Tables.

-The authors should give more details concerning the synthesis and characterization of α -helical PBLG

-The comparison between PBLG-NH₂ (helical)/CE and DBLG(non-helical)/CE is not valid.

-In the caption of Figure 5, “[DBLG]₀ = 30 mM, [NCA]₀ = [CE]₀ = 3 mM” the concentration of DBLG is higher than that of NCA monomer, it must be wrong.

I think that a suitable Journal for this paper is Biomacromolecules or ACS macroLetters, after the authors taking into account the comments and changing the MS accordingly.

Reviewer #2 (Remarks to the Author):

Xia et al. report on the acceleration of amino acid N-carboxyanhydride (NCA) polymerization by crown ethers (or ethylene glycols) in dichloromethane solution. Reaction times are reduced by several orders of magnitude from several hours to few minutes. It is suggested that the rapid polymerization kinetics is due to changed binding geometry between monomer and propagating amine chain end, promoting molecular interaction and lowering activation energy and occurrence of side reactions. The concept is used to prepare a series of well-defined high-molar mass homopolypeptides and block copolypeptides.

This is a very interesting piece of work on a technical high level. However, I feel that the authors need to address and consider some critical points prior to publication.

Maybe this work was inspired by ref. 29, however I do not see the relevance to the amine-initiated NCA polymerization. If the amine was activated by the crown ether (CE), I would have expected that the catalytic activity was lost after the initiation step (formation of rotaxane)?! However, this is not what was observed. Also, any activation of the amine should have resulted in extensive side reactions such as nucleophilic attack of side-chain esters, also not observed. Key seems to be the interaction of CE with NCA (formation of 1:2 complex) and coordination of this complex to the amine chain end (NMR experiments and molecular dynamics simulation). I suggest to draw a scheme of the proposed mechanism for clarification.

Figure 6 (which is difficult to decipher and results are summarized in text) could be moved to SI.

How to rationalize that the catalytic activity of CEs depends on the ring size? Why is the most active ("top-performing") CE the one with medium ring size, i.e., 18-C-6? Maybe molecular dynamics simulations could shed light on this (for CE and LE)? What happens when the CE (or LE) concentration is larger than the active chain end concentration? Is there a point at which acceleration of the polymerization is no longer observed?

How to explain the reaction orders of 0.8 and 0.6 for DBLG and CE, respectively? Why is the reaction order for DBLG in the presence of CE lower than in the absence of CE? In what way does the scheme in Figure SI7a illustrate the CE-catalyzed ring-opening reaction of BLG-NCA by DBLG?

Why all kinetics are plotted as NCA remaining vs. time and not pseudo-first order (i.e., $-\ln(1 - \text{conversion})$ vs. time)?

Why would the initiator efficiency be 15-25% in a two-stage polymerization at quantitative monomer conversion (Table 1)? How do the complete molar mass distributions of these samples look like including the oligomeric fractions (which may not be detected by GPC-LS)? In order to achieve a fast, one-stage polymerization a helical PBLG was used as a macroinitiator. What would happen when a randomly coiled (racemic) PGLB was used? Does the presence of CE also promote an accelerated polymerization of racemic NCAs?

Description of GPC should be included in the main text, methods, including used dn/dc values. GPC chromatograms should also include the RI traces, not only LS traces (which may be different from each other).

The manuscript “Accelerated polymerization of N-carboxyanhydrides catalyzed by crown ether” reports a new synthetic strategy that allows for rapid ring-opening polymerization (ROP) of N-carboxyanhydrides (NCAs) mediated by primary amine and crown ether (CE). Such synthetic protocol is very important to efficiently prepare useful polypeptide materials for differently utilities. The reviewer suggests that the manuscript should be accepted with some revision:

1. The rapid polymerization rate of NCAs under amine and CE is very impressive. In abstract, the author claims “...18-crown-6 enabling the fastest polymerization kinetics ” Here the author should consider comparing this new NCA ROP protocol with other catalysts, including reported Ni and Co complexes (refs. 11-12) and other amine catalysts (trimethylsilyl amines, refs. 15-16) etc. at same monomer feeding ratio and concentration, to demonstrate the fastest reaction rates. A minor question following this one is whether trimethylsilyl amines as initiators with CE will have better controlled ROP or rapid kinetic rates, which is worth discussing.

2. The authors changed CE to amine initiator ratios in the kinetic studies (Fig. 1e) and showed two-stage cooperative polymerization kinetics. However, changing amine to CE ratio could also introduce the possibility that two competing mechanisms (amine-control and CE-mediated amine-control) coexist in the polymerization, rendering difficult to derive kinetic rates. The authors should consider decrease the monomer concentration, and increase monomer to amine ratio, while fixing CE to amine at 1/1, to allow for the measurement of kinetics. Notably, the measurement of kinetic rates and reaction orders are crude (Fig. S7 only three dots to determine the reaction order). More detailed kinetic studies are required to explicitly give the k_{app} value and the reaction orders and of NCA, CE and primary amines. The description about kinetics procedures (SI page S6) is also puzzling as it just measures the initiation step (concentration $[NCA]/[DBLG]/[CE] = 1/10/1$) instead of chain propagation (the equation was given for chain propagation). Additionally, if amine (DBLG) is just initiator and not participates in propagation, the reaction order of 0.79 suggests that DBLG involves in chain propagation, which needs further explanation. The authors should consider fixing CE to NCA ratio (possibly high ratio) with the use of low concentration NCA and adjust amine concentration to see the reaction order of DBLG. In addition, the author should compare the sum of rates of DBLG and CE, with the rate of the ROP reaction at $[DBLG]/[CE]=1/1$, to see if there is any interaction among amine and CE as the coinitiator, or just CE is involved at the chain end for enchainment.

3. The authors have called DCM and chloroform as solvents with low dielectric constants, and said “the polymerization takes much longer time to finish in polar solvents like DMF and tetrahydrofuran (THF)” (page 16). DMF has dielectric constants of 38 and is understandable to behave poorly for the ROP of NCAs. However, THF has lower dielectric constant than DCM (7.5 versus 9.1) and similar dipole moment (1.75 versus 1.6 D). Note that ref. 31 did not discuss and compare DCM or chloroform with THF and DMF. Please explain the solvent effect, or disregard the inaccurate classification.

4. The DOSY and molecular simulation studies only explain that all three molecules involved in the initiation of polymerization studies. Fig 6d-e only showed that DBLG and NCA at opposite side of CE with “enhanced proximity of the ring N-H in BLG-NCA with the α -carbonyl group in DBLG”, which does not reflect the proposed “CE facilitate nucleophilic reactions of primary amines”. In other words, no amine nucleophilic ring-opening reaction route was proposed. Is the amine group on DBLG getting close to C₅-O₁ in NCA ring for regioselective ring-opening reaction, facilitated by CE? More clear explanation about such termolecular complex is needed. Besides, DOSY experiments in Figs. S10-11 not involved the termolecular complex. A macroinitiator amine could be used to replace DBLG to study the termolecular interaction at lower temperature, if the NCA polymerization happened too quickly.

5. Besides, such DOSY and simulation studies only discuss about the initiation. During the chain propagation, is CE staying at the chain end, or instead binding to NCA? Kinetic studies and reaction orders of CE could provide some information. Based on Table 2, using macroinitiator clearly gives better control of the polymerization, indicating that during the propagation, CE could stay at chain end and there is less chain-transfer reaction. On the other hand, the uncontrolled MW and Đ in Table 1 suggest that during the initiation, there are competing chain-transfer reactions, and some initiation species is more active than the other. The termolecular initiation model may need further examination, as it is difficult to exclude two amines with one CE or some other possibilities (oligomer backbiting etc.) during the initiation step (the current simulation is also not convincing). More careful discussion could be better way in the mechanistic studies.

RESPONSES TO REFEREES (NCOMMS-20-07513):

(Reviewer comments in black, our response in blue, and text added to the paper in magenta)

Reviewer #1 (Remarks to the Author):

The authors report a rapid crown ether (CE)-catalyzed ROP of α -amino acid-NCAs in solvents of low dielectric constants, such as DCM, initiated by primary amines. Based on experimental results (kinetics, activation energy, DOSY, binding stoichiometry) and molecular dynamic simulation, they suggest that the presence of CE changes the binding geometry between NCA and propagating amino chain-ends and thus lowers the activation energy, resulting in rapid cooperative ROP.

This MS has a similar philosophy as the two previous papers published by the same groups. In their paper (Nat. Chem. 2017, 9, 614-622, see Figure below), the authors used a TMS-protected amine brush macroinitiator for the ROP of NCAs. They found an auto-accelerated polymerization in the brush-like macromolecular architecture, where the cooperative interactions between the macrodipoles of neighboring helical polypeptides dramatically accelerate the ROP of NCAs. Subsequently, in 2019 (J. Am. Chem. Soc. 2019, 141, 8680-8683, see Figure below), the same groups used linear aliphatic diamines, instead of brush-like macroinitiators, for the ROP of NCAs. They found, similarly, that the “hinged” structure accelerates the ROP dramatically as compared to that initiated by monoamines, such as hexylamine.

We appreciate the comments from the Reviewer. While the previous papers (Nat. Chem. 2017, 9, 614 and J. Am. Chem. Soc. 2019, 141, 8680, as the Reviewer mentioned) and the current manuscript all study the polymerization of NCA in low-polarity solvents like DCM, the novelty and the potential impact of this work are different compared with the previous publications.

The previous work focused on the effect of *macromolecular structure* (i.e., three-dimensional polymer architecture and proximity between initiating sites) on the rate acceleration, which not only help our understanding on cooperative polymerization and protein structure/function, but also potentially facilitate the preparation of polypeptides with complex architecture (e.g., brush-like or star-shaped polypeptides, due to the existence of multiple initiating sites). On the other hand, the current study reported *a ring-opening polymerization catalyst* that promotes the interactions between propagating polypeptide chains and NCA monomers. The CE-promoted rate acceleration does not rely on the specific structure of initiators. Therefore, it is useful for the rapid preparation of polypeptide materials in a broader scope (especially linear polypeptide, which is the most common polymer architecture). The understanding of the catalysis mechanism also guides the future design of new ROP catalysts.

In this MS, the author first used small primary amines to initiate the ROP of NCAs in the presence of crown ethers. The polymerization is fast but uncontrolled, resulting in polypeptides with very high molecular weight than expected and high \bar{D} (1.21-1.33). In order to achieve control over the polymerization, the author used a pre-synthesized helical PBLG (DP=30) as macroinitiator. The authors mention in the introduction, “Nevertheless, the linear polymerization from small molecular primary amines, the conventional initiators for NCA polymerization, is still relatively slow,

especially at low monomer concentrations. Considering that the binding between propagating polypeptide chains and NCA monomers is crucial for the self-catalysis behaviors, we reasoned that a catalyst promoting amine/NCA interactions would further accelerate the polymerization, enabling the rapid and efficient synthesis of polypeptide materials.” However, in the presence of crown ether, the polymerization initiated by commercial available small amines is fast but uncontrolled, resulting in polypeptides with higher molecular weight than the expected ones and high Đ (1.21-1.33).” Consequently, the fast and controlled polymerization can only be achieved by using a pre-synthesized helical PBLG macroinitiator.

We thank the Reviewer for the suggestion. We have corrected our argument in the introduction for clarification by removing the phrase “small molecular”.

The relatively broad MWD of *n*-hexylamine-initiated polypeptides results from the two-stage polymerization kinetics, which is an intrinsic feature of covalent, cooperative polymerization. Since the polymerization rate of the secondary stage is significantly faster than that of the first stage, a small portion of propagating chains quickly consumes the monomers, which is difficult for other chains at the first stage to catch up. This feature was observed in the polymerization without CE and was well-predicted from the kinetic modeling in our previous study (*Nat. Commun.* **2019**, *10*, 5470). Currently, the use of pre-synthesized PBLG macroinitiator is necessary to skip the slow first stage for the simultaneous growth of all polypeptide chains and precise control over MWs. We are still actively seeking a small-molecular primary amine analog to replace PBLG (*e.g.*, a polar amine-containing molecule that resembles the macrodipole of a PBLG macroinitiator to skip the first stage), which will further simplify the system.

The highlight in our argument is the catalysis for NCA polymerization initiated from a simple primary amine, which has a much broader scope than specifically designed catalysts/initiators. For instance, specially designed catalysts/initiators are difficult to be incorporated into the polypeptide synthesis from inorganic NPs or chain extension from other polymers. In addition, our pre-synthesized PBLG macroinitiators can be prepared by various primary amines, which guarantees the functionalization of C-terminus for the resulting polypeptide materials.

A few other comments:

1. I suggest the author to use $\ln\{[M]_0/[M]_t\}$ vs time instead of conversion vs time plots to better analyze the ROP results.

Thank you for the suggestion. We have changed the y-axis from “NCA remaining (%)” to “ $\ln([M]_0/[M]_t)$ ” for all kinetic plots throughout the manuscript for better kinetic analysis.

2. Page 14, “By testing the reaction rate at various concentrations of 18-C-6, we determined that the reaction order is 0.61 in CE (see Supplementary Fig.7 for details) ...” The kinetic study in Figure 7 (SI) is based only on three points. To have more reliable data, more points are needed.

We appreciate the suggestion from the Reviewer. We have re-performed the kinetic study of the ring-opening reaction of BLG-NCA with DBLG, in the presence and absence of 18-C-6, with five points each group for more reliable results. The new results and kinetic analysis were summarized in Supplementary Information (Analysis in Page S7-S8 and kinetic plots in Supplementary Fig. 9,

Page S18, highlighted). Since the released CO₂ was trapped in the closed FTIR cell, which slowed down the polymerization and altered the kinetics (*Macromolecules* **2013**, *46*, 4223), we carried out our new kinetic study in CDCl₃ with ¹H NMR. The overhead space in the NMR tube helped the release of CO₂ for more reliable kinetic results. The experimental condition and reaction order in CE were updated in the main text (Page 15-16, highlighted):

“By testing the reaction rate at various concentration of 18-C-6 in CDCl₃ with NMR, we determined that the reaction order is 0.26 with respect to CE (see Supplementary Fig. 9 for details)”

3. Figure 3C, the GPC plot, corresponding to the polymerization with [M]/[I]=1000 shows a visible shoulder (usually indicating an overlap of two molecular weight distributions). I suggest the author to carry out a polymerization with monomer/initiator ratio between 500 and 1000 to see the gradual changes of molecular weight distribution.

We thank the Reviewer for the comments. We carried out the polymerization at [M]₀/[I]₀ = 750, which showed a monomodal peak on GPC-LS trace and indicated well-controlled polymerization process (*D* = 1.06). The new polymerization results were included in Fig. 3 and Table 2 in the main text (Page 11 and 13, respectively, highlighted).

4. The authors should include the DOSY spectra in Supplementary Information.

Thank you for the suggestion. The DOSY spectra were included as Supplementary Figs. 18-21 (Page S31-S34, highlighted).

5. The authors should include the dn/dc values, used for the molecular weights calculations of the resultant polypeptides, in the note of the Tables.

Thank you for the suggestion. The dn/dc value was added in the footnote of Table 1 and Table 2 in the main text (Page 12 and 13, respectively, highlighted) and Supplementary Table 1 and Table 2 in the Supplementary Information (Page S26 and S27, respectively, highlighted).

6. The authors should give more details concerning the synthesis and characterization of α-helical PBLG.

We appreciate the comment from the Reviewer. The preparation of α-helical PBLG as a macroinitiator was well described in our previous publication (*Nat. Commun.* **2019**, *10*, 5470), and the current study followed the same protocol. The GPC-LS trace of the PBLG macroinitiator was already included as Supplementary Fig. 4a (Supplementary Fig. 6a in the revised manuscript, Page S15). The following text was included in Supplementary Information to briefly discuss the synthesis and characterization of PBLG macroinitiator (Page S2, highlighted):

“Briefly, PBLG-NH₂ was prepared by polymerizing BLG-NCA at 4 °C in DMF with *n*-hexylamine as the initiator ([M]₀/[I]₀ = 30). The resulting PBLG was purified by precipitation in cold hexane/ether (1:1, v/v) and stored at -30 °C. The low temperature during synthesis and storage minimizes the loss of amino groups at the chain terminus. Gel permeation chromatography (GPC) and matrix-assisted laser desorption ionization time-of-flight (MALDI-TOF) MS characterization revealed well-defined polypeptides (see

Supplementary Fig. 6a) with well-reserved end groups (m/z agrees well with calculated value $124.11 + 219.09n$ ($[M+Na]^+$)).”

7. The comparison between PBLG-NH₂ (helical)/CE and DBLG (non-helical)/CE is not valid.

Thank you for the comments. We concur with the Reviewer that DBLG served as a model compound of PBLG-NH₂ in our studies. In a ring-opening reaction of NCA, the rate is dependent on (1) the α -substitutions of amino groups and (2) the secondary structure of the amine-containing propagating chains. The impacts of secondary structure on rate acceleration have been well presented and discussed in our previous publications (*e.g.*, *Nat. Chem.* **2017**, *9*, 614 and *Nat. Commun.* **2019**, *10*, 5470). Therefore, DBLG in the current study serves as a model initiator to mimic the local chemical structure rather than the secondary structure of a propagating polypeptide. Since DBLG has the same α -substitution as a growing PBLG-NH₂ terminus, it provides a satisfactory presentation of the local bulkiness of a propagating polypeptide chain. The sentence below was included in the main text to discuss the selection of DBLG (Page 15, highlighted):

“In order to simplify the study, α,γ -dibenzyl-L-glutamate (DBLG) was selected to mimic the *N*-terminus of polypeptides during the chain propagation step (Fig. 5a), which was used to open the BLG-NCA ring in the presence and absence of CE. DBLG has the same α -substitutions of amino group with the propagating PBLG chain, which serves as an ideal small-molecular compound to mimic the local bulkiness of the propagating polypeptide terminus.”

In addition, for the kinetic study of the ring-opening reaction, the use of polymeric PBLG-NH₂ results in unnecessary difficulties for our analysis: (1) In order to minimize the polymerization and only study the single-step ring-opening reaction, we set high $[-NH_2]_0/[NCA]_0$ ratios (*i.e.*, 5~15). As a result, the large number of protons of PBLG-NH₂ makes it difficult to quantify the signals from NCA protons in ¹H NMR spectra. (2) The use of helical PBLG-NH₂ accelerated the rate of the ring-opening reaction, making it difficult for us to collect enough data points for kinetic analysis. (3) The time and efforts of computer simulation depend on the size of the molecule. The use of large PBLG-NH₂ brings unnecessary elongation of simulation and wastes the valuable computational power.

8. In the caption of Figure 5, “[DBLG]₀ = 30 mM, [NCA]₀ = [CE]₀ = 3 mM” the concentration of DBLG is higher than that of NCA monomer, it must be wrong.

We thank the Reviewer for the comments. The study aims to elucidate the impact of CE on local amine/NCA interaction in a single-step ring-opening reaction rather than a multi-step polymerization. Therefore, we intentionally selected high $[-NH_2]_0/[NCA]_0$ ratios to minimize the polymerization of NCA, which avoids unnecessary complexity for the study, due to the multi-step reactions and the possible change of secondary structures to α -helices. In addition, a high $[-NH_2]_0/[NCA]_0$ ratio was also required for the assumption of pseudo first-order reaction. The following sentence was added in the main text to explain the selection of high $[-NH_2]_0/[NCA]_0$ ratios (Page 15, highlighted):

“The [DBLG]₀/[NCA]₀ ratio was set to 5~15 to minimize the polymerization of NCAs, reducing unnecessary complexity for the analysis.”

I think that a suitable Journal for this paper is Biomacromolecules or ACS macroLetters, after the authors taking into account the comments and changing the MS accordingly.

We appreciate the different opinions from the Reviewer. The novelty of this work is the discovery of an interesting ROP catalyst that promotes the interactions between propagating chains and NCA monomer in low-polarity and low hydrogen-bonding solvents. The simulation studies suggested the change in binding geometry with the addition of CE, which is interesting and provides new insights into the design and mechanistic studies of ROP catalysts.

The simple, stable, metal-free, and commercially available CE catalyst is an excellent complement to the existing library of initiators/catalysts for rapid NCA polymerization (*e.g.*, multi-amine, LiHMDS, PhS-SnMe₃, *N*-heterocyclic carbene, PEG-PBLG emulsion, and fluorinated alcohol). Compared with other catalytic systems, CE works well with various primary amine initiators (which are the most common initiators for NCA polymerization with a large commercially available library), allowing facile functionalization of *C*-terminus of polypeptide materials. In addition, CE provides the rate acceleration for some initiation systems that cannot be easily functionalized or modified, including amine-coated NPs and amine-terminated polymers, which offers great opportunity to prepare polypeptide hybrid materials in a rapid and efficient manner. Moreover, the amine/CE pair is compatible with water, making it an ideal catalytic system to synthesize homopolypeptides from non-purified NCA, according to our recent report (data were already shown in Fig. 4b and Supplementary Fig. 8b).

As a result, we believe our work will draw a broad interest of polymer chemists and materials scientists, and therefore fits the paper quality and journal scope of *Nat. Commun.*

Reviewer #2 (Remarks to the Author):

Xia et al. report on the acceleration of amino acid N-carboxyanhydride (NCA) polymerization by crown ethers (or ethylene glycols) in dichloromethane solution. Reaction times are reduced by several orders of magnitude from several hours to few minutes. It is suggested that the rapid polymerization kinetics is due to changed binding geometry between monomer and propagating amine chain end, promoting molecular interaction and lowering activation energy and occurrence of side reactions. The concept is used to prepare a series of well-defined high-molar mass homopolypeptides and block copolypeptides.

This is a very interesting piece of work on a technical high level.

We thank the Reviewer for the positive comments.

However, I feel that the authors need to address and consider some critical points prior to publication.

1. Maybe this work was inspired by ref. 29, however I do not see the relevance to the amine-initiated NCA polymerization. If the amine was activated by the crown ether (CE), I would have expected that the catalytic activity was lost after the initiation step (formation of rotaxane)?! However, this is not what was observed. Also, any activation of the amine should have resulted in extensive side reactions such as nucleophilic attack of side-chain esters, also not observed. Key seems to be the interaction of CE with NCA (formation of 1:2 complex) and coordination of this complex to the amine chain end (NMR experiments and molecular dynamics simulation). I suggest to draw a scheme of the proposed mechanism for clarification.

We thank the Reviewer for the comments and suggestions. We did not observe any formation of rotaxanes, presumably due to the bulky α -substitution of the amino group at the PBLG chain terminus (please note that the amines used in the rotaxane formation have a less bulky α -substitutions). The resulting PBLG after purification also revealed no proton signals from CE in ^1H NMR spectrum. On the other hand, we indeed observed a similar impact of solvent selection and ring-size on the catalytic effect as Ref. 29 (*J. Am. Chem. Soc.* **2018**, *140*, 6049, Ref. 30 in the revised manuscript). Therefore, we hypothesized the CE promoted the interactions between amines and electrophiles rather than activated the amino groups, which lowered the activation energy and stabilized the transition states of the ring-opening reaction of NCA. We have revised the arguments for clarification (Page 5, highlighted):

“It has been reported that CE facilitated the reactions between primary amines and electrophiles, depending on the solvent selection as well as the size of the ring molecule. Inspired by this result, we hypothesized that CE could also catalyze the NCA polymerization in a proper solvent, which involves the ring-opening reaction of NCA by reacting with amines.”

In addition, we have added a scheme in Supplementary Information (Supplementary Scheme 1, Page S25, highlighted) to elucidate our proposed mechanism of CE-catalyzed polymerization.

2. Figure 6 (which is difficult to decipher and results are summarized in text) could be moved to SI.

We appreciate the Reviewer's comments regarding Figure 6 and do agree that this figure is quite information-dense. It is also true that we do show the primary results presented in Figure 6 in the main text and hope that we have done so in a comprehensible manner that the Reviewer finds clear. We have carefully considered the Reviewer's comments and concur that it may be possible to move Figure 6 to the Supplementary Information, but for two main reasons we respectfully prefer to maintain it in the main text.

First, this is the only figure supporting the results of the molecular simulations and we feel that it is important for conveying these results to the reader that we have a graphical accompaniment in the main text to both provide a visual communication of our results and provide quantitative substantiation of our findings. We appreciate that this manuscript is intended for a diverse audience, but someone from the simulation community reading this work would likely be surprised to see no supporting visual accompaniment to the results articulated in the main text and may feel that

the absence of such a graphic could imply that we were trying to conceal something from the reader.

Second, we do concur that the figure is quite information-dense, but equally feel that it conveys important molecular-level findings. We gave careful consideration in our original submission to the crafting of the figure, its caption, and the main text to make this as comprehensible as possible for the reader. In particular, the caption walks the reader carefully through each of the panels of the figure and explains the essential results pertaining to each. (a) Panel **a** conveys the closer center-of-geometry approach of the molecules and – as detailed in the caption – produces "a 27% enhancement in number density of BLG-NCA/DBLG pairs within a spherical volume of radius $r = 1$ nm". (b,c) Panels **b** and **c** delve deeper into this effect to reveal how CE changes the interaction geometry of the molecules wherein the "most prominent effect is to induce the red sector in the bottom right of the contact matrix to become redder, indicating that the atoms participating in these contacts get closer". (d,e) Panels **d** and **e** provide representative snapshots of interacting DBLG and BLG-NCA molecules in the presence and absence of CE to provide physical illustration of the trends exhibited in the contact matrices. In sum, although the figure requires some scrutiny and effort to comprehend in its totality, we have designed the figure and caption to make this accessible to the non-expert reader as possible, and it contains a wealth of information pertaining to the effect of CE on the proximity and interaction geometry of the DBLG and BLG-NCA that is not experimentally accessible and we feel is very valuable to communicate to the interested reader.

3. How to rationalize that the catalytic activity of CEs depends on the ring size? Why is the most active ("top-performing") CE the one with medium ring size, i.e., 18-C-6? Maybe molecular dynamics simulations could shed light on this (for CE and LE)? What happens when the CE (or LE) concentration is larger than the active chain end concentration? Is there a point at which acceleration of the polymerization is no longer observed?

Thank you for the comments. We attributed the top catalytic performance of 18-C-6 to its suitable conformation to bind with amine/NCA and stabilization of the transition state. Similar ring-size-dependence was also observed in other reports on CE binding and catalysis (e.g., *J. Am. Chem. Soc.* **1993**, *115*, 2837 and *J. Am. Chem. Soc.* **2018**, *140*, 6049).

We concur with the Reviewer that molecular simulation can, and indeed does, provide molecular level understanding of this effect. It was, in fact, the primary intent of the molecular simulations presented in this manuscript and the value of Figure 6 (see the response to Q2) to do just this. In particular, we compared the prevalence and geometry of the pairwise molecular interactions of DBLG and BLG-NCA in the presence and absence of 18-C-6. Due to the high cost of these calculations, we did not consider all sizes of the ring and their linear analogs, but instead performed a head-to-head comparison of these interactions in the absence and presence of the 18-C-6 that was shown experimentally to be top performing. The results of the molecular simulations provided compelling molecular-level support that the properties of 18-C-6 are such that it enhances the prevalence of close DBLG/BLG-NCA interactions and the geometry of these interactions. By comparing to simulations conducted in its absence, these calculations provide supporting evidence that the physical size and physicochemical interactions of the 18-C-6 CE ring induce the formation of termolecular complexes with a potentially favorable geometry for ring-opening reaction of NCA.

In this sense, the molecular simulations accomplished their goal in providing a molecular-level rationalization for the experimentally observed catalytic enhancements associated with 18-C-6.

We do concur that a more comprehensive study geared towards understanding why this *particular* ring size should be optimal may perform additional calculations involving the various other CE and LE structures. This would be a rather expensive computational endeavor but could well be a very worthwhile follow up to better understand the high performance of 18-C-6, particularly if we were to couple this with electronic structure methods (perhaps mixed QM/MM calculations) capable of directly simulating the BLG-NCA/DBLG reaction.

For the second part of the Reviewer's question, we carried out the polymerization of BLG-NCA with high $[CE]_0/[I]_0$ ratios (*i.e.*, 20:1 and 50:1). Both polymerizations proceeded rapidly, with 95% conversion of NCA observed within 25 and 11 min at $[CE]_0/[I]_0 = 20:1$ and 50:1, respectively. Nevertheless, the polymerization rate at $[CE]_0/[I]_0 = 1:1$ was already fast that the change in acceleration at high $[CE]_0/[I]_0$ ratios was not significant. The new kinetic plots were included as Supplementary Fig. S4a (Page S13, highlighted). The sentence below was added in the main text to discuss the impact of $[CE]_0/[I]_0$ on the polymerization kinetics (Page 8, highlighted):

“While the increase in $[CE]_0/[I]_0$ ratio did not significantly change the rate acceleration behavior (Supplementary Fig. 4a), the decrease in $[CE]_0/[I]_0$ to 1:20 slowed down the polymerization kinetics, with 74% conversion after 6 h that is still much faster than that in the absence of CE (Supplementary Fig. 4b).”

4. How to explain the reaction orders of 0.8 and 0.6 for DBLG and CE, respectively? Why is the reaction order for DBLG in the presence of CE lower than in the absence of CE? In what way does the scheme in Figure SI7a illustrate the CE-catalyzed ring-opening reaction of BLG-NCA by DBLG?

We appreciate the interesting questions from the Reviewer. We have re-performed the kinetic study of ring-opening reaction of BLG-NCA with DBLG, in the presence and absence of 18-C-6, with five points each group for more reliable results. The new results and kinetic analysis were summarized in Supplementary Information (Analysis in Page S7-S8 and kinetic plots in Supplementary Fig. 9, Page S18, highlighted). Since the released CO₂ was trapped in the closed FTIR cell, which slowed down the polymerization and altered the kinetics (*Macromolecules* **2013**, *46*, 4223), we carried out our new kinetic study in CDCl₃ with ¹H NMR. The overhead space in the NMR tube helped the release of CO₂ for more reliable kinetic results.

The new reaction order was determined to be 0.50 and 0.26 for DBLG and CE, respectively. The positive reaction order with respect to CE indicates the catalytic role of CE. The reaction order of 0.50 with respect to DBLG, on the other hand, suggests a non-conventional mechanism that deviates from the “normal amine mechanism” with a reaction order of 1.0 with respect to initiator (Kricheldorf, H. R. *α-aminoacid-N-carboxy-anhydrides and related heterocycles: syntheses, properties, peptide synthesis, polymerization*, Springer-Verlag, 1987). In addition, the decrease in the reaction order with respect to DBLG was still observed upon the addition of CE (from 0.64 to 0.50). Nevertheless, the ring-opening reaction of NCA (no matter in the presence or absence of CE) is obviously not an elementary reaction (the non-integral reaction orders also indicate a

complicated mechanism). Therefore, the elucidation of reaction orders requires in-depth calculation of energies of reaction intermediates (*e.g.*, *Macromol. Chem. Phys.* **2010**, *211*, 1708), which falls beyond the scope of the current work and will be covered in our follow-up studies.

The scheme in Supplementary Fig. 7a (Supplementary Fig. 9a in the revised manuscript) illustrates the overall chemical equation of a single-step ring-opening reaction of NCA by DBLG, which mimics the ring-opening reaction during the chain propagation step. We intentionally selected DBLG bearing the same α -substitutions of the amino group compared with a propagating PBLG chain. Therefore, the proposed CE-catalyzed ring-opening mechanism should be similar with that of the CE-catalyzed polymerization, which was already presented in Supplementary Scheme 1 (Page S25, highlighted) as the Reviewer suggested (see the response to Q1).

5. Why all kinetics are plotted as NCA remaining vs. time and not pseudo-first order (i.e., $-\ln(1-\text{conversion})$ vs. time)?

Thank you for the suggestion. We have changed the y-axis from “NCA remaining (%)” to “ $\ln([M]_0/[M]_t)$ ” for all kinetic plots throughout the manuscript for better kinetic analysis.

6. Why would the initiator efficiency be 15-25% in a two-stage polymerization at quantitative monomer conversion (Table 1)? How do the complete molar mass distributions of these samples look like including the oligomeric fractions (which may not be detected by GPC-LS)? In order to achieve a fast, one-stage polymerization a helical PBLG was used as a macroinitiator. What would happen when a randomly coiled (racemic) PGLB was used? Does the presence of CE also promote an accelerated polymerization of racemic NCAs?

We appreciate the insightful comments from the Reviewer. The low apparent initiation efficiency results from the two-stage kinetics with a faster secondary stage, which agrees well with our previously reported systems (*Nat. Chem.* **2017**, *9*, 614 and *Nat. Commun.* **2019**, *10*, 5470). Briefly, a fraction of propagating polypeptide chains entering secondary stage outgrow other chains in the first stage, leading to larger MWs than expected values (*i.e.*, calculated from $[M]_0/[I]_0$). The critical polypeptide chain length at the transition point is ~ 10 . Therefore, the oligomeric species with DP < 10 cannot be effectively detected by GPC (even on the GPC-dRI trace, where the oligomeric species overlap with the solvent peaks). In fact, the kinetic modeling in our previous studies revealed the complete molecular weight distributions containing the low-MW species, and predicted the decrease in the mass percentage of these oligomeric species with increasing $[M]_0$ or $[M]_0/[I]_0$ (*Nat. Commun.* **2019**, *10*, 5470). Nevertheless, the addition of CE introduces parallel reactions with and without the participation of CE as catalysts, which requires a large piece of kinetic data to model the kinetic parameters. Therefore, the kinetic modeling of CE-catalyzed system and the subsequent prediction of complete MWD are out of the scope of the current work and will be included in our future publication.

To answer the second part of the Reviewer’s question, we carried out the CE-catalyzed polymerization of chiral γ -benzyl-D-glutamate NCA (BDG-NCA) and racemic γ -benzyl-DL-glutamate NCA (BDLG-NCA) initiated with *n*-hexylamine. While the polymerization rate of BDG-NCA is comparable with that of BLG-NCA ($> 95\%$ within 28 min), the polymerization of BDLG-NCA is much slower, reaching 50% monomer conversion after 3 h. The addition of CE

still promoted the polymerization of racemic BDLG-NCA, but the catalyzed polymerization was slower than that of BLG-NCA and BDG-NCA. The new polymerization kinetics of BDG-NCA and BDLG-NCA were included as Fig. 1e and Supplementary Fig. S4c (Page 7 and Page S13, respectively, highlighted). The following sentences was added in the main text to discuss the impact of monomer chirality (Page 8, highlighted):

“In addition, while CE catalyzed the polymerization of racemic γ -benzyl-DL-glutamate NCA (BDLG-NCA) (Supplementary Fig. 4c), the catalyzed polymerization was much slower than that of BLG-NCA, with 50% monomer conversion after 3 h (Fig. 1e), suggesting the importance of ordered α -helical structure on the accelerated polymerization. As a comparison, the polymerization of γ -benzyl-D-glutamate NCA (BDG-NCA) showed a comparable rate with that of BLG-NCA (Fig. 1e), indicating a negligible effect of helix sense on the rate enhancement.”

7. Description of GPC should be included in the main text, methods, including used dn/dc values. GPC chromatograms should also include the RI traces, not only LS traces (which may be different from each other).

We thank the Reviewer for the suggestion. The following sentence was included in the methods section in the main text to briefly describe the GPC characterization (Page 21, highlighted):

“The resulting solution was dried under vacuum and injected into GPC without further purification (DMF containing 0.1 M LiBr as the mobile phase).”

The dn/dc value was included in the footnote of Table 1 and Table 2 in the main text (Page 12 and 13, respectively, highlighted) as well as Supplementary Table 1 and Table 2 in the Supplementary Information (Page S26 and S27, respectively, highlighted). In addition, we have provided representative GPC-dRI traces of obtained polypeptide initiated by *n*-hexylamine and PBLG-NH₂ macroinitiator in the presence of CE (Supplementary Fig. S3b and S6c, Page S12 and S15, respectively, highlighted). The GPC-dRI traces revealed monomodal peaks of resulting PBLG from both initiators.

Reviewer #3 (Remarks to the Author):

The manuscript “Accelerated polymerization of *N*-carboxyanhydrides catalyzed by crown ether” reports a new synthetic strategy that allows for rapid ring-opening polymerization (ROP) of *N*-carboxyanhydrides (NCAs) mediated by primary amine and crown ether (CE). Such synthetic protocol is very important to efficiently prepare useful polypeptide materials for differently utilities.

We appreciate the positive comments from the Reviewer.

The reviewer suggests that the manuscript should be accepted with some revision:

1. The rapid polymerization rate of NCAs under amine and CE is very impressive. In abstract, the author claims “...18-crown-6 enabling the fastest polymerization kinetics” Here the author should

consider comparing this new NCA ROP protocol with other catalysts, including reported Ni and Co complexes (refs. 11-12) and other amine catalysts (trimethylsilyl amines, refs. 15-16) etc. at same monomer feeding ratio and concentration, to demonstrate the fastest reaction rates. A minor question following this one is whether trimethylsilyl amines as initiators with CE will have better controlled ROP or rapid kinetic rates, which is worth discussing.

We thank the Reviewer for the insightful suggestion. We carried out the polymerization of BLG-NCA initiated by HMDS and Ni(bipy)(COD) under the same conditions (*i.e.*, DCM as the solvent, $[M]_0 = 50$ mM, $[M]_0/[I]_0 = 100$), both exhibited slower polymerization than the new CE-catalyzed polymerization. HMDS-initiated polymerization reached 90% conversion after 150 min, and the conversion of Ni(bipy)(COD) was only ~ 20% after 6 h (as shown in Fig. R1 below). We attributed the slow polymerization initiated by Ni(bipy)(COD) to the change in solvent and monomer concentration, as previous papers reported the polymerization in THF or DMF at a higher concentration (*e.g.*, *J. Am. Chem. Soc.* **1998**, *120*, 4240 and *J. Am. Chem. Soc.* **2000**, *122*, 5710).

Fig. R1 Semilogarithmic kinetic plot of polymerization of BLG-NCA in DCM initiated by Ni(bipy)(COD). $[M]_0 = 50$ mM, $[M]_0/[I]_0 = 100$.

The catalytic effect of CE on HMDS-initiated polymerization was also evaluated. In the presence of 18-C-6, HMDS-initiated polymerization showed insignificant rate acceleration, presumably due to the bulky *N*-trimethylsilyl carbamate at the chain terminus during polypeptide propagation (*J. Am. Chem. Soc.* **2007**, *129*, 14114), which blocked the interaction of *N*-terminus with CE. In fact, the HMDS-initiated polymerization showed similar conversion in the late stage (*i.e.*, > 80% conversion) in the presence and absence of 18-C-6. This result suggested that the CE catalyzed the NCA polymerization by interacting with the amino group at the propagating chain terminus, which agrees well with our arguments. The HMDS-initiated polymerization in the presence and absence of 18-C-6 was added as Supplementary Fig. 10b (Page S19, highlighted). The following sentence was added in the main text to discuss the catalytic effect of CE on HMDS-initiated polymerization (Page 17, highlighted):

“In addition, the addition of 18-C-6 exhibited insignificant acceleration of hexamethyldisilazane (HMDS)-initiated polymerization of BLG-NCA (Supplementary Fig. 10b), presumably due to the existence of bulky *N*-trimethylsilyl carbamate at the chain terminus during chain propagation, which blocked the interactions with CE.”

2. The authors changed CE to amine initiator ratios in the kinetic studies (Fig. 1e) and showed two-stage cooperative polymerization kinetics. However, changing amine to CE ratio could also introduce the possibility that two competing mechanisms (amine-control and CE-mediated amine-control) coexist in the polymerization, rendering difficult to derive kinetic rates. The authors should consider decrease the monomer concentration, and increase monomer to amine ratio, while fixing CE to amine at 1/1, to allow for the measurement of kinetics. Notably, the measurement of kinetic rates and reaction orders are crude (Fig. S7 only three dots to determine the reaction order). More detailed kinetic studies are required to explicitly give the k_{app} value and the reaction orders and of NCA, CE and primary amines. The description about kinetics procedures (SI page S6) is also puzzling as it just measures the initiation step (concentration $[NCA]/[DBLG]/[CE] = 1/10/1$) instead of chain propagation (the equation was given for chain propagation). Additionally, if amine (DBLG) is just initiator and not participates in propagation, the reaction order of 0.79 suggests that DBLG involves in chain propagation, which needs further explanation. The authors should consider fixing CE to NCA ratio (possibly high ratio) with the use of low concentration NCA and adjust amine concentration to see the reaction order of DBLG. In addition, the author should compare the sum of rates of DBLG and CE, with the rate of the ROP reaction at $[DBLG]/[CE]=1/1$, to see if there is any interaction among amine and CE as the coinitorator, or just CE is involved at the chain end for enchainment.

We appreciate the detailed and helpful suggestions from the Reviewer.

We concur with the Reviewer that two competing mechanisms exist when $[CE]_0/[I]_0 < 1$. To better visualize the first stage, we used circular dichroism (CD) to characterize the secondary structure of the propagating polypeptide at $[CE]_0/[I]_0 = 1$. CD characterization allows us to catch some early time point (as early as 0.5 min). The time plot of CD signal at 222 nm, which is the characteristic signal for the α -helices, revealed a slow increase within the first 2 min, corresponding well with the first stage when the propagating polypeptide stayed as random coils (*Nat. Chem.* **2017**, *9*, 614). The CD results were included as Supplementary Fig. 3c (Page S12, highlighted). The sentence below was added in the main text to discuss the early stage of CE-catalyzed polymerization (Page 8, highlighted):

“The existence of a slow, first stage during CE-catalyzed polymerization was verified by circular dichroism (CD), where a slow increase in CD signal at 222 nm was observed, corresponding to the random-coiled conformation of propagating polypeptides (Supplementary Fig. 3c)”

We have re-performed the kinetic study of ring-opening reaction of BLG-NCA with DBLG, both in the presence and absence of 18-C-6, with five points each group for more reliable results. Since the released CO_2 was trapped in the closed FTIR cell, which slowed down the polymerization and altered the kinetics (*Macromolecules* **2013**, *46*, 4223), we carried out our new kinetic study in $CDCl_3$ with 1H NMR. The overhead space in the NMR tube helped the release of CO_2 for more reliable kinetic results. We followed the Reviewer’s suggestions for the kinetic studies, where we collected the NMR kinetics with varied $[DBLG]_0$ (or $[18-C-6]_0$) while fixing $[BLG-NCA]_0$ and $[18-C-6]_0$ (or $[DBLG]_0$). The new results and kinetic analysis was summarized in Supplementary

Information (Analysis in Page S7-S8, kinetic plots in Supplementary Fig. 9, Page S18, and summary of k_{app} in Supplementary Table 3, Page S 28, highlighted).

The aim of our studies with DBLG is to elucidate the role of amino groups during *chain propagation step* rather than the initiation step. Therefore, the use of DBLG is to mimic the chemical structure of the propagating chain rather than the initiator. Compared with the initiator (*i.e.*, *n*-hexylamine) without any bulky α -substitutions of the amino group, DBLG has a bulky α -substitution that is the same as a growing PBLG-NH₂ terminus, which offers satisfactory presentation of the local bulkiness of a propagating polypeptide chain. In fact, the bulkiness of α -substitutions of the amino group has a significant effect on the rate of ring-opening reaction of NCA. The rate of the initiation step by *n*-hexylamine is so fast that we were not able to reliably measure it by ¹H NMR (*i.e.*, all NCA rings were open within 1.5 min after mixing).

In order to probe the kinetics of a *single-step* ring-opening reaction during chain propagation, we intentionally selected high [-NH₂]₀/[NCA]₀ ratios to minimize the polymerization of NCA. The polymerization introduces unnecessary complexity for the study, due to the multi-step reactions and the possible change of secondary structures to α -helices. In addition, a high [-NH₂]₀/[NCA]₀ ratio was also required for the assumption of pseudo first-order reaction. The following sentence was added in the main text to clarify the role of DBLG and the selection of high [-NH₂]₀/[NCA]₀ ratios (Page 15, highlighted):

“In order to simplify the study, α,γ -dibenzyl-L-glutamate (DBLG) was selected to mimic the *N*-terminus of polypeptides during the chain propagation step (Fig. 5a), which was used to open the BLG-NCA ring in the presence and absence of CE. DBLG has the same α -substitutions of amino group with the propagating PBLG chain, which serves as an ideal small-molecular compound to mimic the local bulkiness of the propagating polypeptide terminus. The [DBLG]₀/[NCA]₀ ratio was set to 5~15 to minimize the polymerization of NCAs, reducing unnecessary complexity for the analysis.”

In order to check whether CE served as a co-initiator during NCA polymerization, we added 18-C-6 at the different stage of *n*-hexylamine-initiated polymerization of BLG-NCA (*i.e.*, before initiation, during the first stage with a random-coiled propagating polypeptide chain, or during the second stage with an α -helical propagating polypeptide chain), all exhibited rate acceleration upon the introduction of 18-C-6. This result excluded the possibility that CE served as the co-initiator. The results were added as Supplementary Fig. 2b (Page S11, highlighted). The sentence below was added in the main text to discuss the result (Page 6, highlighted):

“Additionally, CE was introduced at different time points during the Hex-NH₂-initiated polymerization of BLG-NCA. The monomer was rapidly consumed upon the addition of CE in all cases (Supplementary Fig. 2b), excluding the possibility that CE served as a co-initiator.”

3. The authors have called DCM and chloroform as solvents with low dielectric constants, and said “the polymerization takes much longer time to finish in polar solvents like DMF and tetrahydrofuran (THF)” (page 16). DMF has dielectric constants of 38 and is understandable to behave poorly for the ROP of NCAs. However, THF has lower dielectric constant than DCM (7.5

versus 9.1) and similar dipole moment (1.75 versus 1.6 D). Note that ref. 31 did not discuss and compare DCM or chloroform with THF and DMF. Please explain the solvent effect, or disregard the inaccurate classification.

Thank you for the suggestion. The fast polymerization of BLG-NCA in DCM or chloroform was related to not only the macrodipole of α -helices (*Nat. Chem.* **2017**, *9*, 614), but also the hydrogen-bonding interactions between propagating amino groups and NCA monomers (*Nat. Commun.* **2019**, *10*, 5470). Therefore, the use of solvents with either high polarity or strong hydrogen-bonding ability slows down the NCA polymerization. Ref. 31 (*J. Am. Chem. Soc.* **1976**, *98*, 377, Ref. 32 in the revised manuscript) discussed the β -scale of solvent hydrogen-bond acceptor (HBA) basicity, revealing THF to be a relatively strong hydrogen-bond acceptor ($\beta = 0.523$). DCM, on the other hand, is a non-hydrogen-bonding solvent that is only discussed in Table 1 in Ref. 31, but not used for later calculation of β -scale. We have modified the related arguments for clarification (highlighted):

“Herein we report the use of crown ether (CE) to catalyze the polymerization of NCA initiated by conventional primary amine initiators in solvents with low polarity and low hydrogen-bonding ability.”

“...the polymerization takes much longer time to finish in polar or hydrogen-bonding solvents like DMF and tetrahydrofuran (THF)...”

“In summary, we reported the CE-catalyzed, accelerated polymerization of NCAs in solvents with low polarity and low hydrogen-bonding ability.”

4. The DOSY and molecular simulation studies only explain that all three molecules involved in the initiation of polymerization studies. Fig 6d-e only showed that DBLG and NCA at opposite side of CE with “enhanced proximity of the ring N-H in BLG-NCA with the α -carbonyl group in DBLG”, which does not reflect the proposed “CE facilitate nucleophilic reactions of primary amines”. In other words, no amine nucleophilic ring-opening reaction route was proposed. Is the amine group on DBLG getting close to C5-O1 in NCA ring for regioselective ring-opening reaction, facilitated by CE? More clear explanation about such termolecular complex is needed. Besides, DOSY experiments in Figs. S10-11 not involved the termolecular complex. A macroinitiator amine could be used to replace DBLG to study the termolecular interaction at lower temperature, if the NCA polymerization happened too quickly.

We thank the Reviewer for the comments. We concur with the Reviewer that the DOSY and molecular simulation only suggested that CE promoted the interactions between DBLG (*i.e.*, mimic of propagating polypeptide chain) and NCA. We hypothesized that the enhanced interactions resulted in lower activation energy and stabilization of transition state. However, further explanation of reaction mechanism relies on the in-depth calculation of energies of reaction intermediates (*e.g.*, *Macromol. Chem. Phys.* **2010**, *211*, 1708), which is beyond the scope of the current work and will be studied in our future projects. The region-selective mechanism suggested by the Reviewer, on the other hand, may not be the main mechanism for the rate acceleration, since the 5-C=O ring-opening reaction by primary amine is dominant even in the absence of CE. It has been reported in previous studies that the undesired 2-C=O ring-opening reaction by primary

amine, which led to the formation of ureido acid chain end, was very limited in NCA polymerization with “normal amine mechanism” (0.15%, *J. Am. Chem. Soc.* **1966**, 88, 3627). The argument below was added to the main text to discuss the proposed mechanism (Page 18, highlighted):

“Since the addition of C⁵=O of NCA ring by NH₂ group of propagating amine is the rate-limiting step during NCA polymerization, CE-promoted interactions between NCA and propagating polypeptides likely lower the activation energy and stabilize the transition state, leading to the rate acceleration of ROP.”

The low-temperature DOSY experiment involving the termolecular complex was performed as the Reviewer suggested. After the mixing of BLG-NCA with CE/PBLG-NH₂ mixture, a significant decrease in the diffusion coefficient of BLG-NCA (10%) was observed, suggesting the binding interactions between PBLG-NH₂ and BLG-NCA. The new DOSY results were included as Supplementary Fig. 11e (Page S20, DOSY spectra as Supplementary Fig. 21 in Page S34). The following sentence was included in the main text to discuss the new DOSY results (Page 18, highlighted):

“In an attempt to directly probe the termolecular interactions, we collected the DOSY spectra of CE/PBLG-NH₂/NCA at low temperature to slow down the reaction. Upon the mixing of BLG-NCA with CE/PBLG-NH₂ mixture, a decrease in the diffusion coefficient of BLG-NCA was observed (Supplementary Fig. 11e), validating the interaction of NCA with the PBLG-NH₂ chains.”

5. Besides, such DOSY and simulation studies only discuss about the initiation. During the chain propagation, is CE staying at the chain end, or instead binding to NCA? Kinetic studies and reaction orders of CE could provide some information. Based on Table 2, using macroinitiator clearly gives better control of the polymerization, indicating that during the propagation, CE could stay at chain end and there is less chain-transfer reaction. On the other hand, the uncontrolled MW and Đ in Table 1 suggest that during the initiation, there are competing chain-transfer reactions, and some initiation species is more active than the other. The termolecular initiation model may need further examination, as it is difficult to exclude two amines with one CE or some other possibilities (oligomer backbiting etc.) during the initiation step (the current simulation is also not convincing). More careful discussion could be better way in the mechanistic studies.

We appreciate the comments from the Reviewer.

We want to emphasize that the model ring-opening reaction of BLG-NCA by DBLG mimics the ring-opening reaction during the *chain propagation step* rather than initiation (see the response to Q2). The rate of NCA ring-opening reaction was dependent on the secondary structure and the α -substitutions of amino groups. Since the former factor was well studied in our previous publications (*e.g.*, *Nat. Chem.* **2017**, 9, 614 and *Nat. Commun.* **2019**, 10, 5470), we selected DBLG as a model amine (with the same α -substitutions as a propagating PBLG chain) to represent the local bulkiness of a propagating chain. The initiation step by *n*-hexylamine, on the other hand, is much faster even in the absence of CE due to the less bulky α -substitutions (*i.e.*, two protons). In

addition, we indeed used PBLG-NH₂ rather than *n*-hexylamine for DOSY studies, which elucidated the binding interactions during the chain propagation step instead of initiation step.

Since we have evidences showing that CE can interact with NCA, amino group, as well as NCA/PBLG-NH₂ to form a termolecular complex (Fig. 6 and Supplementary Figs. 10-13 in the revised manuscript), we believe that all three interactions exist during the polymerization of NCA. Nevertheless, only the formation of termolecular complex leads to the ring-opening reaction and the consumption of NCA. CE promoted the interactions between propagating polypeptides and NCA, which lowered the activation energy and stabilized the transition state (see the response to Q4). A scheme illustrating the role of CE in the accelerated polymerization of NCA was proposed and included as Supplementary Scheme 1 (Page S25, highlighted).

The relatively broad MWD of *n*-hexylamine-initiated polypeptides (*i.e.*, data in Table 1) results from the two-stage polymerization kinetics with a faster secondary stage, which is an intrinsic feature of cooperative polymerization. Since the polymerization rate of the secondary stage is significantly faster than that of the first stage, a small portion of propagating chains entering the secondary stage (*i.e.*, forming stable α -helices) quickly consumes the monomers and outgrows the short chains in the first stage (*i.e.*, chains adopting random-coiled conformation). This feature was also observed in the polymerization without CE, and was well-predicted from the kinetic modeling in our previous study (*Nat. Commun.* **2019**, *10*, 5470). As a result, the use of pre-synthesized, α -helical PBLG macroinitiator is necessary to skip the slow first stage for simultaneous growth of all polypeptide chains and precise control over MWs (*i.e.*, data in Table 2).

While we cannot completely rule out the possibility of more competing chain-transfer reactions in the first, random-coiled stage, we have previously shown narrower MWD with increasing $[M]_0$ and $[M]_0/[I]_0$ for *n*-hexylamine-initiated polymerization without CE, mainly due to the “catch-up” of short chains (*Nat. Commun.* **2019**, *10*, 5470). This result suggests that the short chains in the first stage remain active for further chain extensions. Additionally, we have shown previously that the accelerated polymerization in DCM help minimize the side reactions including chain transfers (*ACS Macro Lett.* **2019**, *8*, 1517). Therefore, we believe that the chain transfers play an insignificant role in the broad MWD of *n*-hexylamine-initiated polypeptides. In-depth evaluation of side reactions like chain transfers will be covered in our future studies.

REVIEWER COMMENTS

Reviewer #1 (Remarks to the Author):

The authors successfully responded to all comments of Reviewers 1 and 2 and made all necessary corrections/additions to the revised MS.

The following minor issues need to be addressed before publication:

(1) As the authors claimed that (page 9) "Various functions, including the pyrenyl group (Pyr) as a fluorescence detector and the propargyl group (PP) for post-polymerization modification, can be easily integrated into the C-terminus of the polypeptides by selecting the corresponding primary amines as primers (Supplementary Figure 5c)", I suggest to the authors to perform end-group analysis of these functionalized polypeptides using MALDI-TOF and NMR.

(2) In order to show the generality of their method, I suggest to the authors to study the ROPs using pyrenyl- and propargyl-functionalized oligomeric initiators.

Reviewer #3 (Remarks to the Author):

Just one minor problem about this revised manuscript:

1. For the first question (page 11 in the rebuttal letter), instead of using the working conditions for Ni complex, the authors chose DCM which is not the right solvent for NCA/Ni polymerization. Please perform the kinetic studies in either DMF or THF as those are the working conditions and should give quantitative monomer conversion in hours. The same thing for HMDS, please also perform the reaction in DMF. Please put all three working reactions together (Ni, HMDS, amine/CE, even solvents are different) for kinetic rates comparison.

This reviewer was asked to comment on the the authors' comments to reviewer #2's concerns:

I went through the authors' answers to the reviewer #2's comments. The authors have answered well on question 1 about the mechanism. For the simulation questions in # 2 and 3, I think the authors addressed them well and I agree that the figure 6 should be kept in the main text. For the comments # 4, 5, 6, the authors have re-performed the kinetics studies (which I also asked in my comments) and proved their arguments. The authors added detailed GPC information for comment 7. Therefore, the authors have fully addressed the reviewer #2 's questions and I do not have any additional comments on that part.

RESPONSES TO REFEREES (NCOMMS-20-07513A):

(Reviewer comments in black, our response in blue, and text added to the paper in magenta)

Reviewer #1 (Remarks to the Author):

The authors successfully responded to all comments of Reviewers 1 and 2 and made all necessary corrections/additions to the revised MS.

We appreciate the positive comments from the Reviewer.

The following minor issues need to be addressed before publication:

(1) As the authors claimed that (page 9) “Various functions, including the pyrenyl group (Pyr) as a fluorescence detector and the propargyl group (PP) for post-polymerization modification, can be easily integrated into the C-terminus of the polypeptides by selecting the corresponding primary amines as primers (Supplementary Figure 5c)”, I suggest to the authors to perform end-group analysis of these functionalized polypeptides using MALDI-TOF and NMR.

Thank you for the suggestion. We have performed the end-group analysis of the functionalized polypeptides as the Reviewer suggested. Due to the difficulty in ionizing high-MW polypeptides, we performed a CE-catalyzed polymerization at $[M]_0/[I]_0$ (*i.e.*, 25) and stopped the polymerization at early stage (*i.e.*, 40 s). The MALDI-TOF analysis revealed two series of peaks, with the obtained m/z signals agreeing well with the calculated values of $[M+H]^+$ and $[M+NH_4]^+$, suggesting the successful incorporation of functional pyrenyl and propargyl groups. In addition, the pyrenyl functional groups were also visible from 1H NMR spectra (propargyl group reacted with the polypeptide-essential TFA-*d* solvent that was not suitable for NMR analysis). The end-group analysis was included as Supplementary Fig. 6 (Page S16, highlighted), the following text was added in the main text to highlight the successful C-terminus functionalization (Page 9, highlighted):

“By stopping the polymerization at the early stage, we successfully confirmed the incorporation of these functional groups through NMR and MALDI-TOF MS (Supplementary Fig. 6).”

(2) In order to show the generality of their method, I suggest to the authors to study the ROPs using pyrenyl- and propargyl-functionalized oligomeric initiators.

We appreciate the suggestion from the Reviewer. We have prepared pyrenyl- and propargyl-functionalized PBLG macroinitiators in a similar way with hexylamine-functionalized PBLG macroinitiator, which were then used for the CE-catalyzed polymerization. Both macroinitiators resulted in well-defined polypeptides with predictable MW and narrow dispersity ($D = 1.05$). GPC-UV characterization also confirmed the successful incorporation of pyrenyl groups.

The new results were added as Supplementary Fig. 9 (Page S19, highlighted), the following sentence was added in the main text to discuss the broader scope of the macroinitiator strategy (Page 12-13, highlighted):

“In addition, Pyr-NH₂ and PP-NH₂ were used to prepare C-terminus functionalized PBLG macroinitiators, which enabled the preparation of functionalized polypeptides with predictable MWs in the presence of CE (Supplementary Fig. 9).”

Reviewer #3 (Remarks to the Author):

Just one minor problem about this revised manuscript:

1. For the first question (page 11 in the rebuttal letter), instead of using the working conditions for Ni complex, the authors chose DCM which is not the right solvent for NCA/Ni polymerization. Please perform the kinetic studies in either DMF or THF as those are the working conditions and should give quantitative monomer conversion in hours. The same thing for HMDS, please also perform the reaction in DMF. Please put all three working reactions together (Ni, HMDS, amine/CE, even solvents are different) for kinetic rates comparison.

We thank the Reviewer for the suggestion. We performed the polymerization of BLG-NCA initiated by bipyNi(COD) in THF as well as initiated by HMDS in DMF. Both polymerizations were slower compared with the CE-catalyzed polymerization in DCM, even at a higher [M]₀ (*i.e.*, 200 mM). The new polymerization results were included as Supplementary Fig. 1c (Page S11, highlighted), the following text was added in the main text to discuss the fast polymerization catalyzed by CE compared with other initiating systems (Page 6, highlighted):

“Additionally, with the same batch of NCA, CE-catalyzed polymerization in DCM was much faster than conventional polymerization systems, including bipyNi(COD) initiating system in THF and hexamethyldisilazane (HMDS) initiating system in DMF (Supplementary Fig. 1c).”

This reviewer was asked to comment on the the authors' comments to reviewer #2's concerns:

I went through the authors' answers to the reviewer #2's comments. The authors have answered well on question 1 about the mechanism. For the simulation questions in # 2 and 3, I think the authors addressed them well and I agree that the figure 6 should be kept in the main text. For the comments # 4, 5, 6, the authors have re-performed the kinetics studies (which I also asked in my comments) and proved their arguments. The authors added detailed GPC information for comment 7. Therefore, the authors have fully addressed the reviewer #2 's questions and I do not have any additional comments on that part.

We appreciate the positive comments from the Reviewer.

REVIEWERS' COMMENTS

Reviewer #1 (Remarks to the Author):

The authors successfully responded to all comments of Reviewers 1 and 2, carried out the proposed experiments (MALDI-TOF, NMR of a low molecular weight polypeptide; ROP of NCA using functionalized PBLG macroinitiators) and made all necessary corrections/additions to the revised MS, and SI. I fully support the publication of this work in Nature Communications.

Reviewer #3 (Remarks to the Author):

The authors have successfully address all the questions and profoundly demonstrate the superior reactivity of crown-ether assisted NCA polymerization. Therefore, I suggest the manuscript should be accepted.

RESPONSES TO REFEREES (NCOMMS-20-07513B):

(Reviewer comments in black, and our response in blue)

Reviewer #1 (Remarks to the Author):

The authors successfully responded to all comments of Reviewers 1 and 2, carried out the proposed experiments (MALDI-TOF, NMR of a low molecular weight polypeptide; ROP of NCA using functionalized PBLG macroinitiators) and made all necessary corrections/additions to the revised MS, and SI. I fully support the publication of this work in Nature Communications.

We appreciate the positive comments from the Reviewer. We thank the Reviewer for all comments and suggestions during the paper revision.

Reviewer #3 (Remarks to the Author):

The authors have successfully addressed all the questions and profoundly demonstrate the superior reactivity of crown-ether assisted NCA polymerization. Therefore, I suggest the manuscript should be accepted.

Thank you for the positive comments. We appreciate your comments and suggestions during the paper revision.